# A Theory of Transfer-Based Black-Box Attacks: Explanation and Implications

**Yanbo Chen, Weiwei Liu**[*]
School of Computer Science, Wuhan University
National Engineering Research Center for Multimedia Software, Wuhan University
Institute of Artificial Intelligence, Wuhan University
Hubei Key Laboratory of Multimedia and Network Communication Engineering, Wuhan University
{yanbo.acad,liuweiwei863}@gmail.com

## Abstract

Transfer-based attacks [1] are a practical method of black-box adversarial attacks in which the attacker aims to craft adversarial examples from a source model that is transferable to the target model. Many empirical works [2–6] have tried to explain the transferability of adversarial examples from different angles. However, these works only provide ad hoc explanations without quantitative analyses. The theory behind transfer-based attacks remains a mystery.

This paper studies transfer-based attacks under a unified theoretical framework. We propose an explanatory model, called the *manifold attack model*, that formalizes popular beliefs and explains the existing empirical results. Our model explains why adversarial examples are transferable even when the source model is inaccurate as observed in Papernot et al. [7]. Moreover, our model implies that the existence of transferable adversarial examples depends on the "curvature" of the data manifold, which further explains why the success rates of transfer-based attacks are hard to improve. We also discuss our model's expressive power and applicability.

## 1 Introduction

Machine learning (ML) models are vulnerable to adversarial examples [8, 9]. It has been observed that adversarial examples of one model can fool another model [9, 10], i.e., adversarial examples are transferable. Utilizing this transferability, researchers have developed some practical black-box attack methods, in which they first obtain a source model and then use the adversarial examples of the source model to attack the target model [5, 11]. This type of attack is referred to in the literature as transfer-based attacks (TBA).

TBAs have many intriguing properties. For instance, Papernot et al. [7] find that the adversarial examples of an *inaccurate* source model can transfer to the target model with a non-negligible success rate. Besides, the success rates of TBAs are constantly lower than other methods of black-box attacks [5, 6, 12]; the low success rate seems to be an intrinsic property of TBAs.

Previous works have tried to explain these properties from different perspectives [2–6]. Unfortunately, these works only provide ad hoc explanations for the properties of TBAs without quantitative analyses under a unified framework. For example, Dong et al. [6] suggest that the low success rates of TBAs are due to the adversarial examples of the source model falling into a "non-adversarial region" of the target model, while the properties of such regions are not discussed quantitatively. As for the cause of transferability, a self-evident explanation is that the source and target models have similar decision boundaries [10]. However, these similarities are hard to characterize, and it is even harder to associate

---

[*]Correspondence to: Weiwei Liu <liuweiwei863@gmail.com>.

37th Conference on Neural Information Processing Systems (NeurIPS 2023).

the decision boundary similarity with the properties of TBAs. More importantly, this explanation contradicts the abovementioned experimental results in Papernot et al. [7], i.e., adversarial examples are also transferable when the source and target model does not have similar decision boundaries. We refer the readers to Section 2 for more related works. In summary, existing analyses make good sense under their respective settings, but it is unclear whether they are in accord with each other. It is natural to ask the following question:

*Can we find a unified theoretical framework that explains the properties of TBAs?*

This paper gives a positive answer to this question. More specifically, we propose an explanatory model, called the *manifold attack model*, that formalizes popular beliefs and explains the existing empirical results in a unified theoretical framework. Our model assumes that the natural data lies on a low-dimensional manifold [13–16] denoted by $\mathcal{M}$; see Section 2 for more related works on this assumption. As the central part of our model, we specify a hypothesis class $\mathcal{F}_{\mathcal{M}}$ and assume that both the source and the target model come from $\mathcal{F}_{\mathcal{M}}$. The construction of $\mathcal{F}_{\mathcal{M}}$ is motivated by the following two widespread beliefs. In order to classify unseen data correctly, ML models are believed to have extracted *semantic information* (i.e., the ground truth) of the natural data after training. On the other hand, ML models are also able to learn non-robust features that generalize well in test phrases [17, 18]. These features reflect the *geometrical information* of the training data [17].

In our model, the hypothesis class $\mathcal{F}_{\mathcal{M}}$ is designed to capture both semantic and geometric information of the data. As we will define in Section 4, the classifiers in $\mathcal{F}_{\mathcal{M}}$ can be decomposed into the product of a basis classifier $f_b$ and a multiplier $\phi$, i.e., $f = f_b \cdot \phi$ for $\forall f \in \mathcal{F}_{\mathcal{M}}$. We characterize the semantic information as separated sets $A^1, A^2, \cdots, A^k \subset \mathcal{M}$ (for a $k$-class classification task) and capture this information by letting $f_b$ take different values on these sets. The geometrical information $G$ is interpreted as the "approximated shape" of the data manifold, and we let $\phi$ concentrate around $G$. By multiplying $f_b$ and $\phi$ together, we obtain a classifier that captures the semantic and geometry information of the training data. In brief, our model assumes that both the source and target models come from a specified hypothesis class and the natural data is drawn from a low-dimensional manifold.

Despite its simple form, the manifold attack model provides us with powerful tools to theoretically analyze the properties of TBAs. More specifically, Theorem 4.8 proves that the off-manifold adversarial examples are transferable when the source model is inaccurate, which explains the empirical results from Papernot et al. [7]. By further discussing the existence of off-manifold adversarial examples, we theoretically explain why the success rates of TBAs are hard to improve. Moreover, Theorem 4.13 quantitatively discusses the relationship between the existence of transferable adversarial examples and the "curvature" of the data manifold $\mathcal{M}$, which formalizes the explanation in Dong et al. [6].

In addition to the explanatory results, we further discuss the expressive power and the possible extensions of our model in general applications. Section 5.1 proves that our model is extensible, i.e., we can replace the hypothesis class $\mathcal{F}_{\mathcal{M}}$ by the family of ReLU networks while proving similar results. Our model builds a bridge between the less developed theory behind TBAs and the huge amount of theoretical works analyzing ReLU networks. Furthermore, Section 5.2 demonstrate that the expressive power of our model is strong enough for the study of TBAs. We also provide a detailed discussion on the applicability of our model in Appendix A.

In summary, we propose a model that is theoretically tractable and consistent with existing results. It formalizes widespread beliefs and explains the properties of TBAs under a unified framework. The remainder of this paper is structured as follows. Sections 2 and 3 introduce the related works and terminologies, respectively. We propose our model and present the explanatory results in Section 4. Section 5 makes further discussions on our model. Section 6 summarizes this paper. Some remarks and the omitted proofs can be found in the appendix.

## 2   Related Works

The related works of TBAs and the low-dimensional manifold assumption are summarized as follows.

### 2.1   Transfer-Based Attacks

The research on the adversarial robustness of ML can be roughly divided into adversarial attacks [8, 9, 19], defenses [20–22], and the analyses of the robustness of existing methods [23–26]. Adversarial

attacks can be divided into two classes based on whether the attackers have the gradient information of the target model, i.e., white-box [27] and black-box [7] attacks. TBAs are one of the two main approaches to performing black-box attacks on ML models. Apart from TBAs, the other main approach is the optimization-based attack that approximates the gradient of the target model and performs white-box attacks thereafter [28–30]. While TBAs require much less information from the target model (and thus more practical) than optimize-based attacks, the success rates of TBAs are constantly lower, even when the source models are almost accurate [2, 3, 5, 6, 11].

There are a few theoretical works that have tried to explain the properties of TBAs. However, these works rely heavily on either simple models or strong assumptions. Apart from those mentioned in Section 1, the seminal work of Goodfellow et al. [9] tries to explain the transferability of adversarial examples using linear models. Charles et al. [31] theoretically prove, in the context of linear classifiers and two-layer ReLU networks, that transferable adversarial examples exist. However, [31] does not explain those properties of TBAs mentioned in Section 1. Gilmer et al. [32] assume that the natural data is drawn from a "concentric spheres dataset", which can be viewed as a special form of the low-dimensional data manifold. In comparison to previous works, our model requires milder assumptions on both data distribution and hypothesis class, while providing a more detailed discussion of the properties of TBAs.

### 2.2 The Low-Dimensional Manifold Assumption

The low-dimensional manifold assumption is commonly seen in many areas of research, ranging from classic approximation theory [13, 14] to computer vision [15, 16]. Under this assumption, it is intuitive to divide the adversarial examples into two groups based on whether they lie on or off the data manifold. Both on-manifold [15, 33] and off-manifold [32, 34] adversarial examples have been studied in many previous works. In most cases, a classifier is simultaneously vulnerable to both on- and off-manifold adversarial examples [33].

In this paper, we discuss the transferability of both on- and off-manifold adversarial examples. We are mainly motivated by the techniques from Zhang et al. [35] and Li et al. [14]. The main results of Zhang et al. [35] decompose the adversarial risk based on the position of the adversarial example and discuss the existence of different types of adversarial examples; Zhang et al. [35] provides us with useful tools to analyze the existence of transferable adversarial examples. Li et al. [14] consider robust generalization and approximate continuous classifiers by ReLU networks. In this paper, we prove that the hypothesis class $\mathcal{F}_{\mathcal{M}}$ in our model can also be approximated by ReLU networks.

## 3 Problem Setup

This section introduces the basic setups and terminologies. We use a black triangle sign (▲) to indicate the end of assumptions, definitions, or remarks.

### 3.1 Notations

Let $\mathbb{R}^d$ be the real vector space equipped with the $l_p$-norm, where $p \in [1, +\infty]$. In this paper, we consider the $k$-class classification problems ($k \geq 2$) on $\mathbb{R}^d$. Denote the data-label pairs by $(\mathbf{x}, y)$, in which $y$ is chosen from some label space $\mathcal{Y}$. By convention, we let $\mathcal{Y} = \{-1, 1\}$ in binary classification and $\mathcal{Y} = \{1, 2, \cdots, k\}$ when $k > 2$. Unless otherwise specified, we assume that the natural data lie on a low-dimensional manifold $\mathcal{M} \subset \mathbb{R}^d$. More specifically, we adopt the following assumption.

**Assumption 1** (Low-dimensional Manifold). Let $\mathcal{M} \subset [0, 1]^d$ be a compact smooth manifold[2] and assume that the dimension of $\mathcal{M}$ is less than $d$. Let $D$ be some given continuous distribution that is supported on $\mathcal{M}$. We say that a random variable $\mathbf{x}$ is *natural data* if $\mathbf{x} \sim D$.  ▲

Denote the support set of $D$ by supp($D$). With slight abuse of notation, let $y$ also be a mapping from supp($D$) to $\mathcal{Y}$ that assigns each natural data $\mathbf{x} \sim D$ with a true label $y = y(\mathbf{x})$. Notice that only natural data is endowed with a true label. Since supp($D$) $\subset [0, 1]^d$, we will make no distinction between $\mathbb{R}^d$ and $[0, 1]^d$ in the remainder of this paper without loss of generality (WLOG).

---

[2]Some realistic datasets are supported on $[0, 1]^d$, e.g., MNIST and Fashion-MNIST.

Let $d_p$ be the metric induced by the $l_p$-norm and $B(\mathbf{x}; r)$ the $l_p$-ball around $\mathbf{x}$ with radius $r$. For any subset $S \subset \mathbb{R}^d$, let $d_p(\mathbf{x}, S) := \inf_{\mathbf{x}' \in S} d_p(\mathbf{x}, \mathbf{x}')$ be the distance between $S$ and $\mathbf{x}$. Let the classifiers $f$ be functions defined on $\mathbb{R}^d$. When $k > 2$, we consider $f(\mathbf{x}) = (f^{(1)}(\mathbf{x}), f^{(2)}(\mathbf{x}), \cdots, f^{(k)}(\mathbf{x}))^T$ that maps $\mathbf{x} \in \mathbb{R}^d$ to $f(\mathbf{x}) \in \mathbb{R}^k$. For ease of notation, let $y(f, \mathbf{x}) := \arg\max_{1 \le i \le k} f^{(i)}(\mathbf{x})$ be the output class of $f$ for $\forall \mathbf{x} \in \mathbb{R}^d$. Since $D$ is continuous, $y(f, \mathbf{x})$ take unique value w.p. (with probability) 1 when $\mathbf{x} \sim D$. In binary classification problems, we consider $f : \mathbb{R}^d \to \mathbb{R}$ and $y(f, \mathbf{x}) = \text{sign}(f(\mathbf{x}))$.

Given perturbation radius $\delta$ and natural data $\mathbf{x} \sim D$, we call $\mathbf{x}_a \in B(\mathbf{x}; \delta)$ an *adversarial example* of $f$ at $\mathbf{x}$ if $y(f, \mathbf{x}) \ne y(f, \mathbf{x}_a)$. Given classifier $f$, let the *standard risk* of $f$ be $R_{\text{std}}(f) := \mathbb{P}_{\mathbf{x} \sim D}[y(f, \mathbf{x}) \ne y(\mathbf{x})]$, and the *adversarial risk* of $f$ with regard to (w.r.t.) $\delta$ is defined as

$$R_{\text{adv}}(f; \delta) := \mathbb{P}_{\mathbf{x} \sim D}[\exists \mathbf{x}_a \in B(\mathbf{x}; \delta) \ s.t. \ y(f, \mathbf{x}) \ne y(f, \mathbf{x}_a)]. \tag{1}$$

Since TBA is often against accurate target models in practice, our paper assumes that the standard risk of the target model $f_t$ is $R_{\text{std}}(f_t) = 0$ for the sake of simplicity. Given such accurate $f_t$, the goal of a TBA is to find adversarial examples of $f_t$. Specifically, the attacker needs to obtain a source model $f_s$ and craft adversarial examples of $f_s$ using white-box attacks. Given natural data $\mathbf{x} \sim D$ and an adversarial example $\mathbf{x}_a \in B(\mathbf{x}; \delta)$ of $f_s$ at $\mathbf{x}$, we say that $\mathbf{x}_a$ *transfers* to $f_t$ if $y(f_t, \mathbf{x}) \ne y(f_t, \mathbf{x}_a)$.

### 3.2 The On- and Off-Manifold Adversarial Examples

In this paper, the on- and off-manifold adversarial examples are defined as follows.

**Definition 3.1** (On- and off-manifold adversarial examples). For any given classifier $f$, perturbation radius $\delta$ and natural data $\mathbf{x} \sim D$, let $\mathbf{x}_a \in B(\mathbf{x}; \delta)$ be an adversarial example of $f$ at $\mathbf{x}$. We call $\mathbf{x}_a$ an *on-manifold* adversarial example $\mathbf{x}_a \in \mathcal{M}$, or an *off-manifold* adversarial example if $\mathbf{x}_a \notin \mathcal{M}$.  ▲

We say that a classifier $f$ *suffers from*, or *is vulnerable to* (on-, off-manifold) adversarial examples if $\exists \mathbf{x} \sim D, \mathbf{x}_a \in B(\mathbf{x}; \delta)$ such that $\mathbf{x}_a$ is an (on-, off-manifold) adversarial examples of $f$ at $\mathbf{x}$. If $f$ does not suffer from (on-, off-manifold) adversarial examples, then we consider it to be *robust against* (on-, off-manifold) adversarial examples.

In most cases, the adversarial examples of $f_t$ are regarded as imperceptible to humans (e.g., in a cat-or-dog classification task, the adversarial examples of cat images still look like cats), which implies that adversarial examples are not naturally generated data. Otherwise, the adversarial examples would be endowed with the same label as their corresponding natural data, which leads to a contradiction since $R_{\text{std}}(f_t) = 0$. In other words, adversarial examples should not lie in the support of $D$. To ensure this, we additionally assume that the natural data with different true labels are separated from each other. Notice that there is an intrinsic connection between the natural data $\mathbf{x}$ and its true label $y(\mathbf{x})$, which is often referred to as "semantic information" of the natural data. In this paper, we formalize the *semantic information of the natural data* as a family of sets $\{A^j : j \in \mathcal{Y}\}$, where

$$A^j := \{\mathbf{x} \in \text{supp}(D) : y(\mathbf{x}) = j\}, \ \forall j \in \mathcal{Y}. \tag{2}$$

In practice, the semantic information $A^i$ and $A^j$ is often separated if $i \ne j$ (e.g., Remark A.1). In this paper, we assume that the semantic information is separated in the following sense.

**Assumption 2** (Separated Semantic Information). Given $\lambda > 0$, for $\forall i \ne j \in \mathcal{Y}$, we assume that $A^i$ and $A^j$ are $2\lambda$-separated. Here, for any given subsets $E, F \subset \mathbb{R}^d$ and $r > 0$, we say that $E$ and $F$ are *r-separated* if $d_p(\mathbf{x}_1, \mathbf{x}_2) \ge r$ for $\forall \mathbf{x}_1 \in E$ and $\mathbf{x}_2 \in F$.  ▲

With the help of Assumption 2, it is easy to check that the adversarial examples would not lie in the support set of $D$ if $\delta < \lambda$. In the rest of this paper, we will treat $\delta$ and $\lambda$ as fixed parameters and always assume that $\delta < \lambda$.

## 4 The Manifold Attack Model

Our proposed model specifies a hypothesis class that has two components. The first one is a semantic classifier that tries to capture the semantic information of the natural data. Similar to Equation (2), we call $\{A^j_f : j \in \mathcal{Y}\}$ the *semantic information learned by a classifier* $f$ if $y(f, \mathbf{x}) = j$ for $\forall \mathbf{x} \in A^j_f$ and $\forall j \in \mathcal{Y}$. To avoid trivial discussion, we assume that $A^j_f \ne \emptyset$ for $\forall j \in \mathcal{Y}$. The following definition specifies a family of classifiers that learn separated semantic information.

**Definition 4.1** (Semantic classifier). We call $f_b$ a *semantic classifier* if there is a family of pairwise $2\lambda$-separated set $\{A_f^j : j \in \mathcal{Y}\}$ such that $\{A_f^j : j \in \mathcal{Y}\}$ is the semantic information learned by $f_b$. ▲

It is easy to check that $R_{\text{std}}(f_b) = 0$ if $A^j \subset A_f^j$ for $\forall j \in \mathcal{Y}$. Intuitively, the accuracy of a classifier depends on how well it learns the semantic information of the natural data.

The second component of our model is a function that captures the geometric structure of the data.

**Definition 4.2** (Concentration multiplier). Given $G \subset \mathbb{R}^d$, we call $\phi$ a *concentration multiplier* around $G$ if $\phi(\mathbf{x}) = 1$ for $\forall \mathbf{x} \in G$, and $\phi(\mathbf{x}_1) < \phi(\mathbf{x}_2)$ for $\forall \mathbf{x}_1, \mathbf{x}_2 \in \mathbb{R}^d$ with $d_p(\mathbf{x}_1, G) > d_p(\mathbf{x}_2, G)$. ▲

Intuitively, the geometric information $G$ can be interpreted as the "approximated shape" of the natural data. The concentration multipliers would assign a much lower confidence score to those data points that are outside of or far away from $G$. In the ideal case, $G$ precisely captures the shape of $\mathcal{M}$, which enables $\phi$ to detect the off-manifold data points (i.e., the out-of-distribution (OOD) data [36, 37]).

As mentioned in Section 1, ML models often capture both the semantic and geometrical information of the natural data. Therefore, we assume that the source and target models can be decomposed into the product of a semantic classifier and a concentration multiplier. More specifically, we consider the hypothesis class defined as follows.

**Definition 4.3.** Consider the $k$-class classification problems ($k \geq 2$) on $[0, 1]^d$. Let $\mathcal{F}_b$ and $\Phi$ be a collection of semantic classifiers and concentration multipliers, respectively. When $k > 2$, let $\mathcal{F}_\mathcal{M}$ be the family of all classifiers $f : [0, 1]^d \to \mathbb{R}^k$ that satisfy

$$f^{(j)}(\mathbf{x}) = f_b^{(j)}(\mathbf{x}) \cdot \phi(\mathbf{x}) \text{ for } \forall \mathbf{x} \in [0, 1]^d \text{ and } \forall j \in \mathcal{Y}. \tag{3}$$

In the binary classification case, we let $f(\mathbf{x}) = f_b(\mathbf{x}) \cdot \phi(\mathbf{x})$ for $\forall \mathbf{x} \in [0, 1]^d$. Here, $f_b$ and $\phi$ are chosen from $\mathcal{F}_b$ and $\Phi$, respectively. Since the semantic information is contented in the manifold, we additionally assume that $\cup_{j \in \mathcal{Y}} A_f^j \subset G$, in which $\{A_f^j : j \in \mathcal{Y}\}$ is the semantic information learned by $f_b$ and $G$ is the approximated shape of $\mathcal{M}$ learned by $\phi$. ▲

Definitions 4.1 to 4.3 together with Assumptions 1 and 2 establish the abstract framework for the manifold attack model. This model formalizes how a classifier captures the semantic and geometric information from natural data. It is worth noticing that choices of $f \in \mathcal{F}_\mathcal{M}$ in Definition 4.3 remains largely arbitrary if we choose $\mathcal{F}_b$ (and $\Phi$) to be the family of all possible semantic classifiers (and concentration multiplier) introduced in Definition 4.1 (and Definition 4.2). In fact, for any $f$ that learns $2\lambda$-separated semantic information $\{A_f^j : j \in \mathcal{Y}\}$ and any $\phi$ such that $(f(\mathbf{x}) = 0) \implies (\phi(\mathbf{x}) = 0)$ for $\forall \mathbf{x} \in \mathbb{R}^d$, we can simply let $f_b^{(j)}(\mathbf{x}) = f^{(j)}(\mathbf{x}) \cdot \phi^{-1}(\mathbf{x})$ for $\forall \mathbf{x} \neq 0$. It is easy to check that $f_b$ is a semantic classifier as long as $A_f^j \cap G \neq \emptyset$ for $\forall j \in \mathcal{Y}$.

The generality in Definition 4.3 comes at the expense of theoretical intractability. Take the semantic classifier for example. In Definition 4.1, $f_b$ cannot be determined by the semantic information it learns. Moreover, $f_b$ cannot even be parameterized according to this definition. There is an implicit tradeoff between generality and theoretical tractability: quantitative analysis of specific scenarios might need more delicate definitions. Next, we introduce one specific combination of $f_b$ and $\phi$ to carry on analyzing the properties of TBAs.

For the sake of simplicity, we only present analyses and results in the context of binary classification in Sections 4 and 5. For ease of notation, let $A := \{\mathbf{x} : y(\mathbf{x}) = 1\}$ and $B := \{\mathbf{x} : y(\mathbf{x}) = -1\}$, i.e., the semantic information of binary natural data. The following proposition specifies a sub-family of semantic classifiers:

**Proposition 4.4** (semantic classifier, binary case). *Given $2\lambda$-separated sets $A_f, B_f \subset \mathcal{M}$. Define:*

$$f_b(\mathbf{x}) = f_b(\mathbf{x}; A_f, B_f) := \frac{d_p(\mathbf{x}, B_f) - d_p(\mathbf{x}, A_f)}{d_p(\mathbf{x}, B_f) + d_p(\mathbf{x}, A_f)}. \tag{4}$$

*Then, $f_b$ is a semantic classifier. In particular, we can obtain from Equation (4) that $f_b(\mathbf{x}) > 0$ if $\mathbf{x}$ is closer (w.r.t. $d_p$) to $A_f$ than $B_f$ and $f_b(\mathbf{x}) < 0$ otherwise.*

Let $\mathcal{F}_{\text{bin}}$ the collection of all semantic classifiers defined by Equation (4) for all possible $2\lambda$-separated $A_f, B_f \subset \mathcal{M}$. The following proposition shows that a semantic classifier can be accurate and robust.

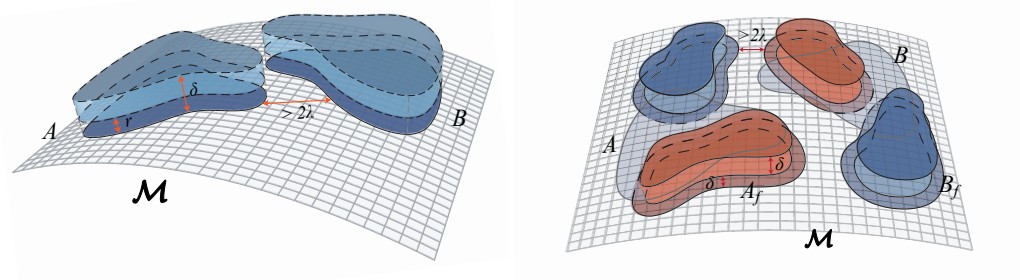

Figure 1: A visualization of the manifold, semantic information, and adversarial examples. The manifold $\mathcal{M}$ is represented by the grid surface. For clarity, we only visualize the above half of $\mathcal{M}$ and omit the below half. **Left**: the $2\lambda$-separated semantic information $A, B \subset \mathcal{M}$ is painted in dark blue. According to Equation (6), the adversarial examples of $f_t = f_b^* \cdot \phi_{\text{off}}$ lie in the shaded bodies in light blue, which is outside of the data manifold. **Right**: the semantic information $A, B$ of the natural data is represented as the light blue area on $\mathcal{M}$. The raised parts upon $\mathcal{M}$ are $\mathcal{N}_\delta(A_f)$ (painted in orange) and $\mathcal{N}_\delta(B_f)$ (painted in dark blue). The intersections of $\mathcal{M}$ and $\mathcal{N}_\delta(A_f)$ (or $\mathcal{N}_\delta(B_f)$) is $A_f$ (or $B_f$), which are indicated by the areas surrounded by the inner contour lines. On the outer contour line of $A_f$ and $B_f$, we have $\phi_{\text{on}} = 0$ since $d_2(\mathbf{x}, \mathcal{N}_\delta(A_f \cup B_f)) = \delta$.

**Proposition 4.5.** *Take $A_f = A$ and $B_f = B$ in Equation* (4) *and denote the corresponding classifier by $f_b^*$. Then, for any given $\lambda \geq \delta > 0$, we have $R_{\text{std}}(f_b^*) = R_{\text{adv}}(f_b^*, \delta) = 0$.*

Similar to Proposition 4.4, we specify a distance-based family of concentration multipliers.

**Proposition 4.6** (Concentration multiplier, binary case). *For any given $r > 0$ and $G \subset \mathbb{R}^d$, denote*

$$\phi(\mathbf{x}) = \phi(\mathbf{x}; r, G) := \frac{r - d_p(\mathbf{x}, G)}{r + d_p(\mathbf{x}, G)}, \quad \forall \mathbf{x} \in \mathbb{R}^d. \tag{5}$$

*Then $\phi(\mathbf{x})$ is a concentration multiplier around $G$.*

Propositions 4.4 and 4.6 introduce a combination of $f_b$ and $\phi$ in binary case. Both $f_b$ and $\phi$ can be parameterized by the semantic and geometrical information it learns. It is also possible to extend our analysis to other scenarios by specifying other combinations of $f_b$ and $\phi$, which is out of the scope of our paper. In the following sections, we adopt the following assumption.

**Assumption 3.** The source and target model are chosen from the hypothesis class given by Definition 4.3, where $\mathcal{F}_b$ and $\Phi$ are the function classes introduced by Propositions 4.4 and 4.6. ▲

With slight abuse of notation, we denote the hypothesis class in Assumption 3 by $\mathcal{F}_\mathcal{M}$. Intuitively, for any $f = f_b \cdot \phi \in \mathcal{F}_\mathcal{M}$, the semantic classifier controls the accuracy of $f$. The similarity between $A, B$ and $A_f, B_f$ reflects how well $f_b$ fits the natural data. Moreover, the overlapping area $(A \cup B) \cap (A_f \cup B_f)$ should contain the training data of $f$ (and thus be non-empty) if $f$ has reached zero training error. The concentration multiplier helps classification by assigning lower confidence scores on those $\mathbf{x} \notin G$, but it might also bring in adversarial examples. Since we only impose mild conditions on the choice of $G$ (i.e., $A_f \cup B_f \subset G$), $\mathcal{F}_\mathcal{M}$ can capture complicated geometrical information. We will come back to the expressive power of $\mathcal{F}_\mathcal{M}$ at Section 5.2.

## 4.1 The Transferability of Adversarial Examples

As a warm-up, we study the robustness of $f \in \mathcal{F}_\mathcal{M}$ against on- and off-manifold adversarial examples without specifying $f_b$ and $\phi$. The following proposition shows that the vulnerability of $f \in \mathcal{F}_\mathcal{M}$ to on- and off-manifold adversarial examples are different.

**Proposition 4.7.** *Let $f = f_b \cdot \phi$. Given $R_{\text{adv}}(f; \delta) \neq 0$, we can obtain (w.p. 1) that*

1. *$f$ suffers from off-manifold adversarial examples.*

2. *if $f_b$ captures the semantic information of the natural data (i.e., $A \subset A_f$ and $B \subset B_f$), then $f$ is robust against on-manifold adversarial examples.*

*Remark.* The results in Proposition 4.7 hold w.p. 1 since our proof ignores some minor events (e.g., $f(\mathbf{x}) = 0$) that occur w.p. 0. For the sake of simplicity, we will not specify those events that happen almost surely in the rest of this paper.

The first result of Proposition 4.7 implies that $f \in \mathcal{F}_{\mathcal{M}}$ suffers from adversarial examples if and only if $f$ suffers from off-manifold adversarial examples. In Section 4.2, we will discuss the existence of transferable adversarial examples based on this result.

We explain the properties of TBAs by proving the existence of $f \in \mathcal{F}_{\mathcal{M}}$ that satisfies desired properties. First of all, we introduce a sub-family of $\mathcal{F}_{\mathcal{M}}$ that is robust against on-manifold adversarial examples. The following concentration multipliers force $f_b$ to concentrate around the data manifold, which brings off-manifold adversarial examples to $f_b$ without introducing on-manifold adversarial examples. For any unspecified $\alpha \in (0, 1)$, let

$$\phi_{\mathrm{off}}(\mathbf{x}) := \phi(\mathbf{x}; \alpha\delta, \mathcal{M}) = \frac{\alpha\delta - d_p(\mathbf{x}, \mathcal{M})}{\alpha\delta + d_p(\mathbf{x}, \mathcal{M})}. \tag{6}$$

Consider the classifiers induced by $\phi_{\mathrm{off}}$, i.e., $f = f_b \cdot \phi_{\mathrm{off}}$. It is clear from Equation (6) that $f(\mathbf{x}) = f_b(\mathbf{x})$ when $\mathbf{x} \in \mathcal{M}$. Combining this with the separated assumptions of semantic information, we can see that $\phi_{\mathrm{off}}$ blocks the on-manifold adversarial examples. Figure 1 demonstrates that the target model $f_t = f_b^* \cdot \phi_{\mathrm{off}}$ is only vulnerable to off-manifold perturbations. The following proposition shows that off-manifold adversarial examples are transferable even if the source model is inaccurate, which explains the phenomenon in Papernot et al. [7].

**Theorem 4.8.** *Consider TBAs with perturbation radius $\delta \in (0, \lambda]$, target model $f_t = f_b^* \cdot \phi_{\mathrm{off}}$ and source model $f_s = f_b \cdot \phi_{\mathrm{off}}$, $f_b \in \mathcal{F}_b$. Denote the semantic information of $f_b$ by $A_f$ and $B_f$. Then, for $\forall \mathbf{x} \in A \cup B$, all adversarial examples (if exist) of $f_s$ at $\mathbf{x}$ are transferable if $\mathbf{x} \in A_f \cup B_f$.*

Theorem 4.8 proves that all of the adversarial examples of $f_s$ are transferable if the concentration multipliers of $f_t$ and $f_s$ are identical, given that $(A \cup B) \cap (A_f \cup B_f) \neq \emptyset$. In particular, $f_s$ is not necessarily accurate, which is consistent with the empirical results in Papernot et al. [7]. Theorem 4.8 also implies that the vulnerability of ML models might be due to the non-robust geometrical information. Such vulnerability is transferable between models that learn similar geometrical information.

Next, we turn to the on-manifold adversarial examples. Proposition 4.7 shows that we cannot find an $f \in \mathcal{F}_{\mathcal{M}}$ such that $f$ suffers from only on-manifold adversarial examples. Instead, we specify a family of concentration multipliers that blocks out off-manifold adversarial examples that is "directly above" the manifold. In this case, we consider Euclidean space $\mathbb{R}^d$ with $l_2$-norm and inner product $\langle \mathbf{x}_1, \mathbf{x}_2 \rangle := \mathbf{x}_1^T \mathbf{x}_2$. For $\forall \mathbf{x} \in \mathcal{M}$, let $T_{\mathbf{x}}\mathcal{M} \subset \mathbb{R}^d$ be the *tangent space* of $\mathcal{M}$ at $\mathbf{x}$, i.e., the space spanned by the possible tangent directions that pass through $\mathbf{x}$. Now that we are considering an inner product space, let $N_{\mathbf{x}}\mathcal{M} \subset \mathbb{R}^d$ be the *normal space* at $\mathbf{x}$ such that $\forall \mathbf{u} \in N_{\mathbf{x}}\mathcal{M}$ and $\mathbf{v} \in T_{\mathbf{x}}\mathcal{M}$, we have $\langle \mathbf{u}, \mathbf{v} \rangle = 0$.

Denote $\mathcal{N}_r(S) := \{\mathbf{x}' \in \mathbb{R}^d : \exists \mathbf{x} \in S \ s.t. \ d_2(\mathbf{x}, \mathbf{x}') < r, \ \mathbf{x}' - \mathbf{x} \in N_{\mathbf{x}}\mathcal{M}\}$ for any given $r > 0$ and $S \subset \mathcal{M}$. We call $\mathcal{N}_r(S)$ a *tubular neighborhood* of $S$ if for $\forall \mathbf{x}' \in \mathcal{N}_r(S)$, there is an unique $\mathbf{x} \in S$ such that $\mathbf{x}' - \mathbf{x} \in N_{\mathbf{x}}\mathcal{M}$. For any semantic classifier $f_b$ with semantic information $A_f$ and $B_f$, define

$$\phi_{\mathrm{on}}(\mathbf{x}) = \phi_{\mathrm{on}}(\mathbf{x}; f_b) := \phi(\mathbf{x}; \delta, \mathcal{N}_\delta(A_f \cup B_f)) \tag{7}$$

and consider $f = f_b \cdot \phi_{\mathrm{on}}(\cdot; f_b)$. According to Equation (7), $\phi_{\mathrm{on}}$ blocks the off-manifold adversarial examples of $f$ that is "directly above" $A_f$ and $B_f$. We use Figure 1 to visualize our idea. The following proposition provides a sufficient condition for adversarial examples that are not transferable.

**Proposition 4.9.** *Consider TBA with perturbation radius $\delta \in (0, \lambda]$, target model $f_t = f_b^* \cdot \phi_{\mathrm{on}}(\cdot, f_b^*)$ and source model $f_s = f_b \cdot \phi_{\mathrm{on}}(\cdot; f_b)$, $f_b \in \mathcal{F}_b$. Denote*

$$S_{\mathrm{crt}} := (A \cap A_f) \cup (B \cap B_f), \ S_{\mathrm{wrg}} := (A \cap B_f) \cup (B \cap A_f). \tag{8}$$

*Then, for $\forall \mathbf{x} \in A \cup B$, we have*

1. *if $B(\mathbf{x}, \delta) \cap \mathcal{M} \subset S_{\mathrm{crt}} \cup S_{\mathrm{wrg}}$, then $f_t$ and $f_s$ are both robust against adversarial examples;*

2. *if $B(\mathbf{x}, \delta) \cap \mathcal{M} \subset A \cup B$ and $\mathbf{x} \in A_f \cup B_f$, then the adversarial examples of $f_s$ at $\mathbf{x}$ (if exists) cannot transfer to $f_t$.*

We interpret $S_{\mathrm{crt}}$ (or $S_{\mathrm{wrg}}$) as the "*correct (or wrong) semantic information*" captured by $f_b$. When $S_{\mathrm{crt}} = \mathrm{supp}(D)$ and $S_{\mathrm{wrg}} = \emptyset$ (i.e., $f_b$ captures the semantic information of the natural data), the

first result of Proposition 4.9 implies that a large part of $\mathbf{x} \in \text{supp}(D)$ is robust against adversarial examples if we block the off-manifold adversarial examples that are "directly above" the semantic information. The second result of Proposition 4.9 implies that the potentially transferable adversarial examples are mostly located inside of the following set

$$(A \cup B) \cap (S_{\text{wrg}} \cup S_{\text{crt}})^c = (A \cup B) \cap (A_f \cup B_f)^c, \tag{9}$$

or at least close to its boundary. Here, we interpret $(A \cup B) \cap (A_f \cup B_f)^c$ as the semantic information not contained in the training data.

In summary, the results in this section together provide a general view of the role that on- and off-manifold adversarial examples play in TBAs. Although both on- and off-manifold adversarial exist, ML models seem to be more vulnerable to off-manifold adversarial examples, and the off-manifold adversarial examples seem to play a more important role in TBAs.

## 4.2 The Existence of Transferable Adversarial Examples

Dong et al. [2] argue that the low success rates of TBAs are possibly due to the adversarial examples of $f_s$ falling into a "non-adversarial region" of $f_t$. In this section, we try to formalize this explanation and study the properties of such non-adversarial regions.

According to Proposition 4.7, $f \in \mathcal{F}_\mathcal{M}$ suffers from adversarial examples if and only if $f$ suffers from off-manifold adversarial examples. This section is devoted to discussing the possible non-existence of off-manifold examples. Before we delve into this problem, let us introduce the notion of robust radius that is initially studied in the certified robustness problems [38, 39].

**Definition 4.10** (Robust Radius, binary case). For any classifier $f$, the *robust radius* of $f$ is defined as the minimum $r \geq 0$ such that for $\forall \mathbf{x} \in A \cup B$ and $\mathbf{x}_1, \mathbf{x}_2 \in B(\mathbf{x}, r)$, we have $f(\mathbf{x}_1)f(\mathbf{x}_2) \geq 0$.

Denote the robust radius of $f$ by $r_\delta(f)$. In our model, the robust radius of $f \in \mathcal{F}_\mathcal{M}$ is controlled by $\phi$. Based on this notion, the following example demonstrates how the existence of off-manifold adversarial examples depends on the shape of the data manifold $\mathcal{M}$.

**Example 4.11.** We use the same setting as Proposition 4.9 and consider the target model $f_t = f_b^* \cdot \phi_{\text{off}}$. Given $0 < r_1 < r_2 < \delta < r_3$, denote $\phi_1 = \phi_{\text{off}}(\cdot; r_1, \mathcal{M})$ and $\phi_2 = \phi_{\text{off}}(\cdot; r_2, \mathcal{M})$. Using Equation (6), we obtain that $r_\delta(f_b^* \cdot \phi_1) = r_1 < \delta$ and $r_\delta(f_b^* \cdot \phi_2) = r_2 < \delta$. However, $r_\delta(f) < \delta$ does not imply that all $\mathbf{x} \in A \cup B$ are vulnerable to adversarial examples.

We demonstrated our idea in Figure A.1. Denote the dark blue surface in Figure A.1 by $\mathcal{M}_{r_2} := \{\mathbf{x} : d_2(\mathbf{x}, \mathcal{M}) = r_2\}$. Observe that $r_2$ is of the same magnitude as the "curvature" of $\mathcal{M}$, we can find a $\mathbf{x}_0 \in A \cup B$ such that $d_2(\mathbf{x}, \mathcal{M}_{r_2}) = r_3 > \delta$, i.e., $\mathbf{x}_0$ is not vulnerable to adversarial examples. Note that all of such $\mathbf{x}_0$ together form the non-adversarial region in Dong et al. [2]. ▲

Example 4.11 shows that the existence of the adversarial examples of $f = f_b \cdot \phi_{\text{off}}$ depends on the "curvature" of $\mathcal{M}$, which seems to be a rather agnostic result. However, we can quantify the "curvature" of $\mathcal{M}$ by the following lemma.

**Lemma 4.12** (Bredon [40]). *Let $\mathcal{M} \subset \mathbb{R}^d$ be a compact smooth manifold, then there is a $\Delta > 0$ such that $N_\Delta \mathcal{M}$ is a tubular neighborhood of $\mathcal{M}$.*

Lemma 4.12 can be viewed as a variant of the tubular neighborhood theorem (cf. Bredon [40]). The constant $\Delta$ is decided by $\mathcal{M}$ and can be used to evaluate the "curvature" of $\mathcal{M}$ in Example 4.11. More specifically, the following proposition provides a sufficient condition for the existence of the off-manifold adversarial examples.

**Theorem 4.13.** *Given perturbation radius $\delta \in (0, \lambda]$ and target model $f_t = f_b^* \cdot \phi_{\text{off}}(\cdot; r, \mathcal{M})$. Let $\Delta$ be the constant specified in Lemma 4.12. Then, for $\forall \mathbf{x} \in A \cup B$, the off-manifold adversarial example of $f_t$ at $\mathbf{x}$ exists if $r < \Delta$.*

Theorem 4.13 establishes a quantitative relationship between the existence of off-manifold examples and the "curvature" of the manifold.

# 5 Discussion

## 5.1 Approximation with ReLU Networks

In applications, the target and source models of TBAs are mostly neural networks, e.g., ReLU networks. In this subsection, we use ReLU networks (cf. Definition 5.1) to approximate the classifiers $f \in \mathcal{F}_\mathcal{M}$. Moreover, we recover some of the results in Section 4 based on the approximated classifier.

We start by introducing the family of ReLU networks. Denote the *rectified linear unit (ReLU)* by $\rho(\mathbf{x}) := \max\{\mathbf{x}, \mathbf{0}\}$, where $\mathbf{0}$ is the zero vector in $\mathbb{R}^d$ and $\max\{\cdot, \cdot\}$ takes the entry-wise maximum of $\mathbf{x}$ and $\mathbf{0}$. The family of deep ReLU networks defined as follows.

**Definition 5.1** (Deep ReLU networks). For any bias vector $\mathbf{b} = (b_1, \cdots, b_L)$ and sequence of weight matrices $\mathbf{W} = (\mathbf{W}_1, \cdots, \mathbf{W}_L)$, we call function $\tilde{f} : \mathbb{R}^d \to \mathbb{R}$ a *(deep) ReLU networks* parametrized by $\mathbf{b}$ and $\mathbf{W}$ if

$$\tilde{f}(\mathbf{x}) = b_L + \mathbf{W}_L \rho(b_{L-1} + \mathbf{W}_{L-1} \rho(\cdots \rho(b_1 + \mathbf{W}_1 \mathbf{x}) \cdots))) \tag{10}$$

for $\forall \mathbf{x} \in \mathbb{R}^d$. Let $W_i$ be a matrix with $m_i$ rows and $n_i$ columns. We have $n_1 = d$, $m_L = 1$, and $m_i = n_{i+1}$ ($1 \le i \le L - 1$).

The non-constant entries in $\mathbf{b}$ and $\mathbf{W}$ are called the *parameters* of $\tilde{f}$. We say that a function is *induced by ReLU networks* if it can be expressed by the linear combination of finite many ReLU networks.

It has been proven [14, 41–43] that ReLU networks can approximate (w.r.t. the sup-norm) continuous functions with a bounded domain to any precision. Here, we denote the sup-norm of a function $f : S \subset \mathbb{R}^d \to \mathbb{R}$ by $\|f\|_\infty := \sup_{\mathbf{x} \in S} |f(\mathbf{x})|$. Specifically, we have:

**Lemma 5.2** (Li et al. [14]). *Given l-Lipschitz function $f : [0, 1]^d \to [-1, 1]$ and precision $\epsilon > 0$, there is a ReLU network $\tilde{f}$ with $O((l/\epsilon)^d) \cdot O(d^2 + d \log(1/\epsilon))$ parameters that satisfies $\|f - \tilde{f}\|_\infty \le \epsilon$.*

Lemma 5.2 provides a useful tool to approximate the Lipschitz functions from $[0, 1]^d$ to $[-1, 1]$. It is straightforward to check the Lipschitzness of $f \in \mathcal{F}_\mathcal{M}$. As a result, given $\epsilon > 0$ and $f \in \mathcal{F}_\mathcal{M}$, we can find an ReLU network $\tilde{f}$ such that $\|f - \tilde{f}\|_\infty < \epsilon$. Unfortunately, $\tilde{f}$ can no longer be decomposed into the product of $f_b$ and $\phi$, which invalidates most of the results in Section 4. We have to make a detour to approximate $f \in \mathcal{F}_\mathcal{M}$. As a first step, we approximate $f_b$ and $\phi$ by $\tilde{f}_b$ and $\tilde{\phi}$, respectively.

**Corollary 5.3.** *Let $f_b$ be a semantic classifier with semantic information $A_f$ and $B_f$ that satisfy a $2\lambda$-separated property. Given $\epsilon > 0$, there is a ReLU network $\tilde{f}_b$ with $O((1/\lambda\epsilon)^d) \cdot O(d^2 + d \log(1/\epsilon))$ parameters such that $\|f_b - \tilde{f}_b\|_\infty \le \epsilon$.*

**Corollary 5.4.** *Given $\epsilon > 0$, $r > 0$ and $S \subset [0, 1]^d$, there is a ReLU network $\tilde{\phi}$ with $O((1/r\epsilon)^d) \cdot O(d^2 + d \log(1/\epsilon))$ parameters that can approximate $\phi(\cdot; r, S)$ to precision $\epsilon$.*

Corollaries 5.3 and 5.4 approximate $f_b$ and $\phi$ with ReLU networks. It remains to approximate the product of two ReLU networks using the following lemma.

**Lemma 5.5** (Yarotsky [41]). *There is a function $\tilde{\times} : [-1, 1]^2 \to [-1, 1]$ induced by ReLU network with $O(\log^2(\epsilon^{-1}))$ parameters such that $\tilde{\times}(x, y) = 0$ if $xy = 0$, and*

$$\sup_{x,y \in [0,1]} |\tilde{\times}(x, y) - xy| \le \epsilon. \tag{11}$$

The above lemma constructs a ReLU network that efficiently approximates the multiplying operator in $\mathbb{R}^2$. The following proposition shows that we can approximate $f \in \mathcal{F}_\mathcal{M}$ with ReLU networks to any precision.

**Proposition 5.6.** *Given $\epsilon, \lambda, \delta, r > 0$, for any $f \in \mathcal{F}_\mathcal{M}$, there is a ReLU network $\tilde{f}$ with*

$$O(\max\{\frac{1}{\lambda\epsilon}, \frac{1}{r\epsilon}\}^d) \cdot O(d^2 + d \log(\frac{1}{\epsilon})) + O(\log^2(\frac{1}{\epsilon})) \tag{12}$$

*parameters that satisfies $\|f - \tilde{f}\|_\infty \le \epsilon$.*

The following theorem partially recovers the results in Theorem 4.8 with ReLU networks.

**Theorem 5.7.** *Consider TBAs with perturbation radius $\delta \in (0, \lambda/2)$, target model $f_t = f_b^* \cdot \phi_{\text{off}}$ and source model $f_s = f_b \cdot \phi_{\text{off}}$, $f_b \in \mathcal{F}_b$. Denote the semantic information of $f_b$ by $A_f$ and $B_f$. Given $\epsilon \le 0.1$, let $\tilde{f}_t$ and $\tilde{f}_s$ be ReLU networks that satisfy*

$$\|\tilde{f}_t - f_t\|_\infty \le \epsilon, \|\tilde{f}_s - f_s\|_\infty \le \epsilon. \tag{13}$$

*Then, for $\forall \mathbf{x} \in (A \cup B) \cap (A_f \cup B_f)$, the adversarial examples $\mathbf{x}_a$ (if exist) of $\tilde{f}_s$ satisfies*

$$\tilde{f}_t(\mathbf{x}) \cdot \tilde{f}_t(\mathbf{x}_a) \leq 2\epsilon(1 + \epsilon)^2 + 2\epsilon^2. \tag{14}$$

Notice that Equation (14) does not imply that the adversarial examples are transferable. Instead, it can only reduce the confidence in the decision made by $\tilde{f}_t$. The results in Theorem 4.8 cannot be fully recovered. When $f$ is close to zero, whether the approximated ReLU network $\tilde{f}$ is greater or less than zero is hard to decide.

The following theorem recovers Proposition 4.9 after modifying some parameters.

**Theorem 5.8.** *Consider TBAs with perturbation radius $\delta \in (0, \lambda/2]$, target model $f_t = f_b^* \cdot \phi_{\mathrm{off}}$ and source model $f_s = f_b \cdot \phi_{\mathrm{off}}$, $f_b \in \mathcal{F}_b$. Denote the semantic information of $f_b$ by $A_f$ and $B_f$. Given $\epsilon \leq 0.1$, let $\tilde{f}_t$ and $\tilde{f}_s$ be ReLU networks that satisfy Equation (13). Then, for $\forall \mathbf{x} \in A \cup B$, we have*

1. *if $B(\mathbf{x}, \delta) \cap \mathcal{M} \subset S_{\mathrm{crt}} \cup S_{\mathrm{wrg}}$, then $\tilde{f}_t$ and $\tilde{f}_s$ are both robust against adversarial examples;*

2. *if $B(\mathbf{x}, \delta) \cap \mathcal{M} \subset A \cup B$ and $\mathbf{x} \in A_f \cup B_f$, then the adversarial examples of $\tilde{f}_s$ at $\mathbf{x}$ (if exists) cannot transfer to $\tilde{f}_t$.*

By approximating the classifiers in $\mathcal{F}_{\mathcal{M}}$, we build a bridge between the less-developed theory behind TBAs and the huge amount of theoretical works analyzing ReLU networks.

## 5.2 The Expressiveness of the Manifold Attack Model

In learning theory, the expressive power of a hypothesis class $\mathcal{F}$ evaluates the performance of $f \in \mathcal{F}$ in fitting natural data; see [44] for a comprehensive overview. Previous works have tried to explain the transferability based on less expressive hypothesis classes, e.g., linear classifiers [32] and the linear combination of "features" [17]. In this subsection, we study the expressive power of our model.

The main goal of TBAs is not to fit the natural data. Instead, for any well-trained classifier $f^* : [0, 1]^d \to \{-1, 1\}$, i.e., $R_{\mathrm{std}}(f^*) = 0$, the goal of our model is to find $f \in \mathcal{F}_{\mathcal{M}}$ such that the adversarial examples of $f$ fits those of $f^*$. More specifically, we have

**Proposition 5.9.** *For any classifier $f^*$ with $R_{\mathrm{std}}(f^*) = 0$ and perturbation radius $\delta \in (r_\delta(f^*), \lambda)$, there is $f \in \mathcal{F}_{\mathcal{M}}$ such that 1) $R_{\mathrm{std}}(f) = R_{\mathrm{std}}(f^*)$, and 2) for $\forall \mathbf{x} \in A \cup B$, if $\mathbf{x}_a$ is an adversarial example of $f^*$ at $\mathbf{x}$, then exists $\mathbf{x}_a' \in B(\mathbf{x}_a, r_\delta(f^*)/4)$ such that $\mathbf{x}_a'$ is an adversarial example of $f$.*

The above proposition implies that our proposed model can cover a wide range of classifiers and is thus sufficient for the study of TBAs on manifolds.

# 6 Conclusion

This paper explains the properties of TBAs under a unified theoretical framework. We suggest that off-manifold adversarial examples play a major role in TBAs. In particular, we show that off-manifold adversarial examples are transferable when the source model is inaccurate. We also prove that the non-existence of off-manifold adversarial examples is one of the reasons why the success rates of TBAs are hard to improve.

## Acknowledgments and Disclosure of Funding

This work is supported by the National Natural Science Foundation of China under Grant 61976161, and the Fundamental Research Funds for the Central Universities under Grant 2042022rc0016.

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

# A Further Discussions

The following remark provides an example of the semantic information of a dataset.

*Remark* A.1 (The semantic information of CIFAR-10). Generally speaking, "semantic" refers to the relationship between natural data and their true label, which should be consistent with human recognition. For example, the semantic information contained in the CIFAR-10 dataset is the true labels (e.g., airplane, automobile, and bird) and their corresponding natural images (e.g., images of airliners, SUVs, and chickens). In this example, an image cannot simultaneously include an airplane and an automobile since "the classes are completely mutually exclusive" in the CIFAR-10 dataset (cf. the official website of CIFAR-10.) ▲

It is also easy to check that the semantic information provided by CIFAR-10 is separated.

## A.1 Visualization of the Non-Adversarial Region

We provide a visualization of Example 4.11 in Figure A.1.

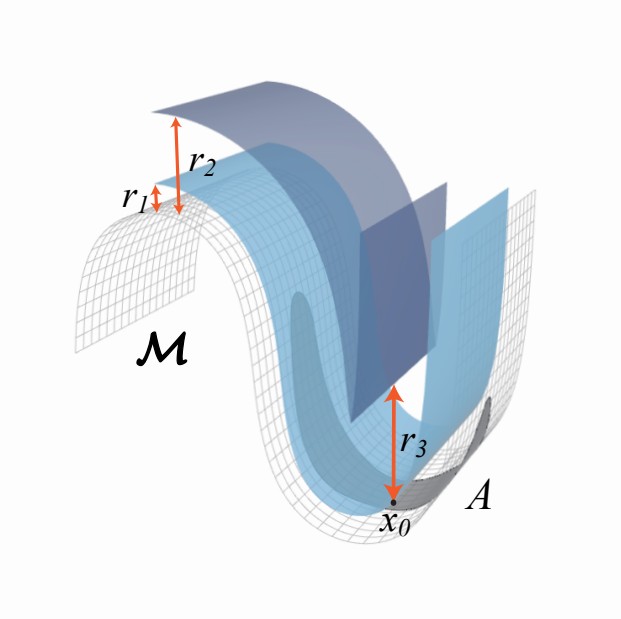

Figure A.1: A visualization of Example 4.11. The data manifold $\mathcal{M}$ is represented by the grid surface. Let the surface in light blue (or dark blue) be the contour surface that $\phi_1 = 0$ (or $\phi_2 = 0$). The distance between $\mathbf{x}_0$ and the dark blue surface is $r_3$, which is greater than $\delta$ and $r_2$. In this visualized example, $x_0$ is robust against off-manifold adversarial examples.

## A.2 The Applicability of the Manifold Attack Model

Recall that Assumption 3 asserts that the classifiers $f \in \mathcal{F}$ can be decomposed into the product of a semantic classifier $f_b$ and a concentration multiplier $\phi$. In this section, we empirically certify the following weaker version of Assumption 3.

**Assumption 4** (Manifold attack model, weaker version). Let $f$ be the source or target model of a TBA. Let $\mathcal{M}_1$ be a compact smooth manifold that is not necessarily a subset of $\mathcal{M}$. Let $\mathcal{N}_{\Delta_1}\mathcal{M}_1$ be a tubular neighborhood of $\mathcal{M}_1$ for some $\Delta_1 > 0$. We assume that for $\forall \mathbf{x} \in \mathcal{M}_1$ and for any non-zero $\boldsymbol{u} \in N_{\mathbf{x}}\mathcal{M}_1$, we have $f(\mathbf{x} + r_1\boldsymbol{u}) > f(\mathbf{x} + r_2\boldsymbol{u})$ holds for $\forall \delta_1 > r_2 > r_1 > 0$.

By letting $G = \mathcal{M}_1$ in Definition 4.2, the classifier in Assumption 3 would also satisfy Assumption 4. In this sense, we claim that Assumption 4 is a weaker version of Assumption 3. Next, we validate Assumption 4 on low-dimensional linearly separated data defined as follows. Given $d > \kappa > 0$,

consider the $\kappa$-dimensional manifold (in this case, a flat plane)

$$\mathcal{M} = \{(x_1, x_2, \cdots, \mathbf{x}_k, 0, \cdots, 0) : x_i \in \mathbb{R}, \forall i = 1, 2, \cdots, \kappa\} \subset \mathbb{R}^d. \tag{A.15}$$

Assume that the natural data $\mathbf{x} \in \mathcal{M}$ is drawn from some continuous distribution $D(\mathbf{x})$, say a mixture of Gaussian distribution. For the sake of simplicity, let the natural data be linearly separable in the following sense. Given $\lambda > 0$ and natural data $\mathbf{x} = (x_1, \cdots, x_d)$, we assume that $|x_1| > \lambda$, and the label of $\mathbf{x}$ is decided by $y(\mathbf{x}) = 1$ if $x_1 > \lambda$, and $y(\mathbf{x}) = -1$ if $x_1 < -\lambda$.

In order to test Assumption 4, our goal is to train a classifier that satisfies this assumption. The training setting is stated as follows. Given $N > 0$, let the training set $S_N = \{\mathbf{x}_1, \mathbf{x}_2, \cdots, \mathbf{x}_N\} \sim D^N(\mathbf{x})$ be independent and identically distributed (i.i.d.) random variables. For $\forall 1 \leq n \leq N$, denote the label of $\mathbf{x}_n$ by $y_N$. Since $D$ is continuous, we assume WLOG that

$$(\exists a \in \mathbb{R} \ s.t. \ \mathbf{x}_i = a\mathbf{x}_j) \implies i = j \tag{A.16}$$

holds for $\forall 1 \leq i, j \leq N$. The hypothesis class is defined as $\mathcal{F}_\mathbf{w} := \{f_\mathbf{w} : \mathbf{w} \in \mathbb{R}^d\}$, where

$$f_\mathbf{w} = \langle \mathbf{w}, \frac{\mathbf{x}}{\|\mathbf{x}\|_2} \rangle \tag{A.17}$$

for $\forall \mathbf{x} \in \mathbb{R}^d$. Consider the logistic loss function $l(u) := \log(1 + \exp(-u))$, $\forall u \in \mathbb{R}$, and the gradient descent (GD) iteration

$$\mathbf{w}(t + 1) = \mathbf{w}(t) - \eta \nabla_\mathbf{w} \mathcal{L}_N(\mathbf{w}(t)). \tag{A.18}$$

We can show (both empirically and theoretically) that:

- The GD iteration converges to an accurate classifier $\hat{f}$,

- $\exists \mathcal{M}_1$ such that $\hat{f}$ satisfies Assumption 4,

which implies that we can actually train an accurate classifier (instead of constructing one using oracle information) that satisfies our assumption. The following section briefly introduces the setting and results of the experiment. The theoretical analysis is omitted.

### A.2.1 Numerical Experiments

In order to visualize our results, the manifold $\mathcal{M}$ is set to be a 2-dimensional plane embedded in $\mathbb{R}^3$. The distribution $D$ is a mixture of two Gaussian distributions with means $(2, 2, 0)$ and $(-2, -2, 0)$, and covariance $\sigma = 0.5$. The optimizer is stochastic gradient descent (SGD). We train a two-layer network, denoted by $\hat{f}$, that reaches 100% testing accuracy. We check whether $\hat{f}$ satisfies Assumption 4. The experimental results are demonstrated in Figure A.2.

**Hardware Specification and Environment** Our experiments are conducted on an Ubuntu 64-bit Linux workstation, having a 10-core Intel Xeon Silver CPU (2.20 GHz) and 4 Nvidia GeForce RTX 2080 Ti GPUs with 11GB graphics memory.

## B Complete Proofs

**Proposition 4.4** (semantic classifier, binary case)**.** *Given $2\lambda$-separated sets $A_f, B_f \subset \mathcal{M}$. Define:*

$$f_b(\mathbf{x}) = f_b(\mathbf{x}; A_f, B_f) := \frac{d_p(\mathbf{x}, B_f) - d_p(\mathbf{x}, A_f)}{d_p(\mathbf{x}, B_f) + d_p(\mathbf{x}, A_f)}. \tag{B.19}$$

*Then, $f_b$ is a semantic classifier. In particular, we can obtain from Equation (4) that $f_b(\mathbf{x}) > 0$ if $\mathbf{x}$ is closer (w.r.t. $d_p$) to $A_f$ than $B_f$ and $f_b(\mathbf{x}) < 0$ otherwise.*

*Proof of Proposition 4.4.* It is easy to check that $f_b(\mathbf{x}) = 1$ when $\mathbf{x} \in A_f$ and $f_b(\mathbf{x}) = -1$ when $\mathbf{x} \in B_f$. By definition, we know that $f_b$ is a semantic classifier. □

**Proposition 4.5.** *Take $A_f = A$ and $B_f = B$ in Equation (4) and denote the corresponding classifier by $f_b^*$. Then, for any given $\lambda \geq \delta > 0$, we have $R_{\mathrm{std}}(f_b^*) = R_{\mathrm{adv}}(f_b^*, \delta) = 0$.*

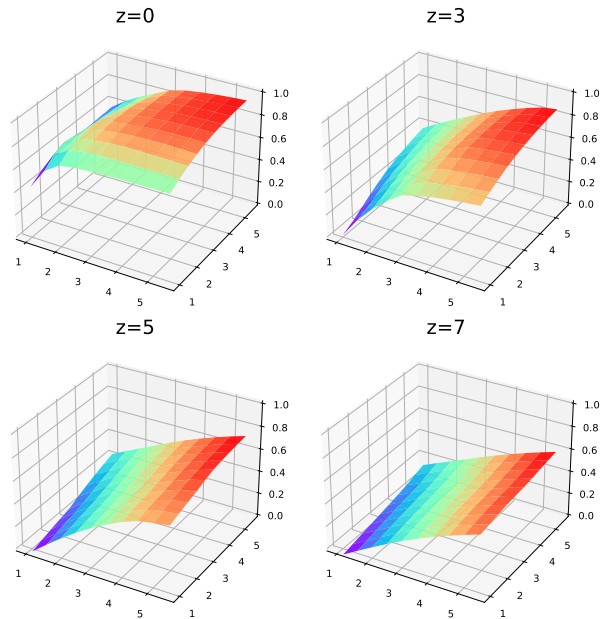

Figure A.2: The experimental results can be interpreted as follows. Denote the vectors $\mathbf{x} \in \mathbb{R}^3$ by $\mathbf{x} = (x, y, z)$. The bottom plane is the manifold, i.e., the $(x, y)$-plane. For each sub-figure, we consider the score of $\hat{f}$ at $(x, y, z)$ for a specific $z$ chosen from $\{0, 3, 5, 7\}$. The height of the surface is the score of $\hat{f}$. Obviously, the score of $\hat{f}$ decreases as $z$ goes larger, which is in accord with the claim in Assumption 4.

*Proof of Proposition 4.5.* By Equation (4), we have

$$f_b^*(\mathbf{x}) = \frac{d_p(\mathbf{x}, B) - d_p(\mathbf{x}, A)}{d_p(\mathbf{x}, B) + d_p(\mathbf{x}, A)}, \ \forall \mathbf{x} \in \mathbb{R}^d. \tag{B.20}$$

Clearly, we have $f_b^*(\mathbf{x}) = 1$ when $\mathbf{x} \in A$ and $f_b^*(\mathbf{x}) = -1$ when $\mathbf{x} \in B$. Then, the standard risk of $f_b^*$ w.r.t. $D(\mathbf{x})$ is

$$R_{\text{std}}(f_b^*) = \mathbb{P}_D \left[ f_b^*(\mathbf{x})y < 0 \mid \mathbf{x} \in A \right] + \mathbb{P}_D \left[ f_b^*(\mathbf{x})y < 0 \mid x \in B \right] = 0 \tag{B.21}$$

Recall that $A$ and $B$ are $2\lambda$-separated (cf. Assumption 2). For $\forall \mathbf{x} \in A$ and $\mathbf{x}' \in B(\mathbf{x}, \delta)$, we have $d_p(\mathbf{x}, B) > \delta$, which implies that $d_p(\mathbf{x}', B) - d_p(\mathbf{x}', A) > 0$, and thus $f_b^*(\mathbf{x}')f_b^*(\mathbf{x}) = f_b^*(\mathbf{x}') > 0$. For $\forall \mathbf{x} \in B$, a similar deduction shows that $f_b^*(\mathbf{x}')f_b^*(\mathbf{x}) > 0$ holds for $\forall \mathbf{x}' \in B(\mathbf{x}, \delta)$. Together, we have

$$R_{\text{adv}}(f_b^*, \delta) := \mathbb{P} \left[ \exists \mathbf{x}' \in B(\mathbf{x}; \delta) \ s.t. \ f_b^*(\mathbf{x}')f_b^*(\mathbf{x}) < 0 \mid \mathbf{x} \in A \right]$$
$$+ \mathbb{P} \left[ \exists \mathbf{x}' \in B(\mathbf{x}; \delta) \ s.t. \ f_b^*(\mathbf{x}')f_b^*(\mathbf{x}) < 0 \mid \mathbf{x} \in B \right] = 0, \tag{B.22}$$

which completes the proof. □

*Remark* B.1. The construction of Equation (B.20) can be found in previous works [14, 45]. In particular, Li et al. [14] uses the ReLU-approximation of $f_b^*$ to study the robust generalization of deep neural networks.

**Proposition 4.6** (Concentration multiplier, binary case). *For any given $r > 0$ and $G \subset \mathbb{R}^d$, denote*

$$\phi(\mathbf{x}) = \phi(\mathbf{x}; r, G) := \frac{r - d_p(\mathbf{x}, G)}{r + d_p(\mathbf{x}, G)}, \ \forall \mathbf{x} \in \mathbb{R}^d. \tag{B.23}$$

*Then $\phi(\mathbf{x})$ is a concentration multiplier around $G$.*

*Proof of Proposition 4.6.* For $\forall \mathbf{x} \in G$, we have $d_p(\mathbf{x}, G) = 0$. That is, $\phi(\mathbf{x}) = 1$ for $\forall \mathbf{x} \in G$. For $\forall \mathbf{x}_1, \mathbf{x}_2$ s.t. $d_p(\mathbf{x}_1, G) > d_p(\mathbf{x}_2, G)$, it is easy to check that $\phi(\mathbf{x}_1) < \phi(\mathbf{x}_2)$. □

**Proposition 4.7.** *Let* $f = f_b \cdot \phi$. *Given* $R_{\text{adv}}(f; \delta) \neq 0$, *we can obtain (w.p. 1) that*

1. $f$ *suffers from off-manifold adversarial examples.*

2. *if* $f_b$ *captures the semantic information of the natural data (i.e., $A \subset A_f$ and $B \subset B_f$), then $f$ is robust against on-manifold adversarial examples.*

*Proof of Proposition 4.7.* We first prove the first result. By definition, there are $r > 0$ and $G \subset \mathbb{R}^d$ such that

$$f(\mathbf{x}) = \frac{d_p(\mathbf{x}, B_f) - d_p(\mathbf{x}, A_f)}{d_p(\mathbf{x}, B_f) + d_p(\mathbf{x}, A_f)} \cdot \frac{r - d_p(\mathbf{x}, G)}{r + d_p(\mathbf{x}, G)}. \tag{B.24}$$

Given $R_{\text{adv}}(f; \delta) \neq 0$, there is natural data $\mathbf{x} \in \text{supp}(D)$ and $x_0 \in B(\mathbf{x}, \delta)$ such that $f(\mathbf{x})f(\mathbf{x}_0) < 0$. If $x_0 \in \mathcal{M}^c$, there is nothing to prove.

Otherwise, we have $\mathbf{x}_0 \in \mathcal{M}$. Denote

$$r_0 := \frac{1}{2} \min\{|d_p(\mathbf{x}_0, A_f) - d_p(\mathbf{x}_0, B_f)|, |r - d_p(\mathbf{x}_0, G)|, |\delta - d_p(\mathbf{x}, \mathbf{x}_0)|\}. \tag{B.25}$$

Consider the non-empty set

$$B(\mathbf{x}, \delta) \cap B(\mathbf{x}_0, r_0) \cap \mathcal{M}^c.$$

For $\forall \mathbf{x}_0' \in B(\mathbf{x}_0, r_0)$, there is

$$d_p(\mathbf{x}, \mathbf{x}_0') \leq d_p(\mathbf{x}, \mathbf{x}_0) + d_p(\mathbf{x}_0, \mathbf{x}_0') < \delta. \tag{B.26}$$

It is also easy to check that

$$\begin{aligned}
\left(d_p(\mathbf{x}_0, B_f) - d_p(\mathbf{x}_0, A_f)\right) \cdot \left(d_p(\mathbf{x}_0', B_f) - d_p(\mathbf{x}_0', A_f)\right) &> 0, \\
\left(r - d_p(\mathbf{x}_0, G)\right) \cdot \left(r - d_p(\mathbf{x}_0', G))\right) &> 0,
\end{aligned} \tag{B.27}$$

which implies that $f(\mathbf{x})f(\mathbf{x}_0') < 0$ and $\mathbf{x}_0'$ is an off-manifold adversarial example.

As for the second result, since $A \subset A_f$, $B \subset B_f$, and $A_f \cup B_f \subset G$, we have

$$\frac{r - d_p(\mathbf{u}, G)}{r + d_p(\mathbf{u}, G)} = 1, \text{ where } \mathbf{u} \in \{\mathbf{x}, \mathbf{x}'\} \tag{B.28}$$

for $\forall \mathbf{x} \in A \cup B$ and $\mathbf{x}' \in B(\mathbf{x}, \delta) \cap \mathcal{M}$. We can easily obtain that $f_b(\mathbf{x}) = f_b(\mathbf{x}')$, which implies that $f$ has no on-manifold adversarial examples. □

**Theorem 4.8.** *Consider TBAs with perturbation radius $\delta \in (0, \lambda]$, target model $f_t = f_b^* \cdot \phi_{\text{off}}$ and source model $f_s = f_b \cdot \phi_{\text{off}}$, $f_b \in \mathcal{F}_b$. Denote the semantic information of $f_b$ by $A_f$ and $B_f$. Then, for $\forall \mathbf{x} \in A \cup B$, all adversarial examples (if exist) of $f_s$ at $\mathbf{x}$ are transferable if $\mathbf{x} \in A_f \cup B_f$.*

*Proof of Theorem 4.8.* If $f_s$ is robust against adversarial examples, then there is nothing to prove. Otherwise, consider $\mathbf{x} \in A_f \cap A$ WLOG. By Equation (6), denote

$$f_s(\mathbf{x}) = f_b(\mathbf{x}) \cdot \phi_{\text{off}}(\mathbf{x}) = \frac{d_p(\mathbf{x}, B_f) - d_p(\mathbf{x}, A_f)}{d_p(\mathbf{x}, B_f) + d_p(\mathbf{x}, A_f)} \cdot \frac{\alpha\delta - d_p(\mathbf{x}, \mathcal{M})}{\alpha\delta + d_p(\mathbf{x}, \mathcal{M})}. \tag{B.29}$$

We can find an adversarial example of $f_s$ at $\mathbf{x}$. Denote this adversarial example by $\mathbf{x}_a$, and we have $f_s(\mathbf{x}_a) < 0$. It is not hard to verify that $d_p(\mathbf{x}_a, B_f) - d_p(\mathbf{x}_a, A_f) > 0$ since $\delta < \lambda$, which implies that $f_b(\mathbf{x}_a) > 0$. It is easy to obtain that $f_s(\mathbf{x}_a) < 0$ and $\phi_{\text{off}}(\mathbf{x}_a) < 0$. We thus have $\alpha\delta < d_p(\mathbf{x}_a, \mathcal{M})$, which implies that $\mathbf{x}_a$ is off the manifold and the distance between $\mathbf{x}_a$ and $\mathcal{M}$ is greater than $\alpha\delta$. In particular, we have

$$\phi_{\text{off}}(\mathbf{x})\phi_{\text{off}}(\mathbf{x}_a) < 0, \tag{B.30}$$

which is independent of the choice of $f_b$. Now consider $f_t(\mathbf{x})$ and $f_t(\mathbf{x}_a)$, where

$$f_t(\mathbf{x}) = f_b^*(\mathbf{x}) \cdot \phi_{\text{off}}(\mathbf{x}) = \frac{d_p(\mathbf{x}, B) - d_p(\mathbf{x}, A)}{d_p(\mathbf{x}, B) + d_p(\mathbf{x}, A)} \cdot \frac{\alpha\delta - d_p(\mathbf{x}, \mathcal{M})}{\alpha\delta + d_p(\mathbf{x}, \mathcal{M})}. \tag{B.31}$$

No matter $\mathbf{x} \in A$ or $\mathbf{x} \in B$, we have for $\forall \mathbf{x}' \in B(\mathbf{x}, \delta)$, there is $f_b^*(\mathbf{x}) = f_b^*(\mathbf{x}')$ (by the $2\lambda$-separated property of $A$ and $B$). By Equation (B.30), we have

$$f_t(\mathbf{x})f_t(\mathbf{x}_a) = f_b^*(\mathbf{x})f_b^*(\mathbf{x}_a) \cdot \phi_{\text{off}}(\mathbf{x})\phi_{\text{off}}(\mathbf{x}_a) < 0, \tag{B.32}$$

i.e., $\mathbf{x}_a$ transfers to $f_t$, which completes the proof. □

**Proposition 4.9.** *Consider TBA with perturbation radius $\delta \in (0, \lambda]$, target model $f_t = f_b^* \cdot \phi_{\mathrm{on}}(\cdot, f_b^*)$ and source model $f_s = f_b \cdot \phi_{\mathrm{on}}(\cdot, f_b)$, $f_b \in \mathcal{F}_b$. Denote*

$$S_{\mathrm{crt}} := (A \cap A_f) \cup (B \cap B_f), \ S_{\mathrm{wrg}} := (A \cap B_f) \cup (B \cap A_f). \tag{B.33}$$

*Then, for $\forall \mathbf{x} \in A \cup B$, we have*

1. *if $B(\mathbf{x}, \delta) \cap \mathcal{M} \subset S_{\mathrm{crt}} \cup S_{\mathrm{wrg}}$, then $f_t$ and $f_s$ are both robust against adversarial examples;*

2. *if $B(\mathbf{x}, \delta) \cap \mathcal{M} \subset A \cup B$ and $\mathbf{x} \in A_f \cup B_f$, then the adversarial examples of $f_s$ at $\mathbf{x}$ (if exists) cannot transfer to $f_t$.*

*Proof of Proposition 4.9.* For the first results, it is easy to check that $\phi_{\mathrm{on}}(\mathbf{x}) = 1$ and $\phi_{\mathrm{on}}(\mathbf{x}_a) = 1$ for $\forall \mathbf{x}_a \in B(\mathbf{x}, \delta)$, which implies that $f$ is robust against adversarial examples. It remains to prove the second result. By Equation (7), denote

$$f_s(\mathbf{x}) = f_b(\mathbf{x}) \cdot \phi_{\mathrm{on}}(\mathbf{x}; f_b) = \frac{d_2(\mathbf{x}, B_f) - d_2(\mathbf{x}, A_f)}{d_2(\mathbf{x}, B_f) + d_2(\mathbf{x}, A_f)} \cdot \frac{\alpha\delta - d_2(\mathbf{x}, \mathcal{N}_\delta(A_f \cup B_f))}{\alpha\delta + d_2(\mathbf{x}, \mathcal{N}_\delta(A_f \cup B_f))}. \tag{B.34}$$

For $\forall \mathbf{x} \in A \cup B$ such that $B(\mathbf{x}, \delta) \cap \mathcal{M} \subset A \cup B$ and $\mathbf{x} \in A_f \cup B_f$, denote the unspecific adversarial example (if exist) of $f_s$ at $\mathbf{x}$ by $\mathbf{x}_a$. Assume that $\mathbf{x} \in A \cap B_f$ WLOG. By definition, we have

$$f_b^*(\mathbf{x}) = f_t(\mathbf{x}) > 0, \ f_b(\mathbf{x}) = f_s(\mathbf{x}) < 0. \tag{B.35}$$

By the $2\lambda$-separated assumption of $A_f$ and $B_f$, we have

$$f_b(\mathbf{x}_a) < 0. \tag{B.36}$$

From $f_s(\mathbf{x}) f_s(\mathbf{x}_a) < 0$, we can obtain that $\phi_{\mathrm{on}}(\mathbf{x}; f_b) \phi_{\mathrm{on}}(\mathbf{x}_a; f_b) < 0$. Since $\mathbf{x} \in B_f$, we have

$$d_2(\mathbf{x}, \mathcal{N}_\delta(A_f \cup B_f)) = 0, \tag{B.37}$$

i.e., $\phi_{\mathrm{on}}(\mathbf{x}; f_b) = 1$. Combine this with $\phi_{\mathrm{on}}(\mathbf{x}; f_b) \phi_{\mathrm{on}}(\mathbf{x}_a; f_b) < 0$, we have $\phi_{\mathrm{on}}(\mathbf{x}_a; f_b) < 0$, i.e.,

$$\alpha\delta < d_2(\mathbf{x}_a, \mathcal{N}_\delta(A_f \cup B_f)) < d_2(\mathbf{x}_a, \mathbf{x}) \le \delta. \tag{B.38}$$

It remains to show that $f_t(\mathbf{x}_a) > 0$. By Equation (7), denote

$$f_t(\mathbf{x}) = f_b^*(\mathbf{x}) \cdot \phi_{\mathrm{on}}(\mathbf{x}; f_b^*) = \frac{d_2(\mathbf{x}, B) - d_2(\mathbf{x}, A)}{d_2(\mathbf{x}, B) + d_2(\mathbf{x}, A)} \cdot \frac{\alpha\delta - d_2(\mathbf{x}, \mathcal{N}_\delta(A \cup B))}{\alpha\delta + d_2(\mathbf{x}, \mathcal{N}_\delta(A \cup B))}. \tag{B.39}$$

By $\mathbf{x} \in A$ and the $2\lambda$-separated assumption of $A$ and $B$, we have $f_b^*(\mathbf{x}_a) > 0$. By $B(\mathbf{x}, \delta) \cap \mathcal{M} \subset A$, we have $\mathbf{x}_a \in \mathcal{N}_\delta(A \cup B))$, i.e., $\phi_{\mathrm{on}}(\mathbf{x}_a; f_b^*) > 0$. Together, we have $f_t(\mathbf{x}_a) = f_b^*(\mathbf{x}_a) \cdot \phi_{\mathrm{on}}(\mathbf{x}_a; f_b^*) > 0$. That is, $\mathbf{x}_a$ is not an adversarial example of $f_t$, which completes the proof. □

**Theorem 4.13.** *Given perturbation radius $\delta \in (0, \lambda]$ and target model $f_t = f_b^* \cdot \phi_{\mathrm{off}}(\cdot; r, \mathcal{M})$. Let $\Delta$ be the constant specified in Lemma 4.12. Then, for $\forall \mathbf{x} \in A \cup B$, the off-manifold adversarial example of $f_t$ at $\mathbf{x}$ exists if $r < \Delta$.*

*Proof of Theorem 4.13.* For $\forall \mathbf{x} \in A \cup B$, let $\mathbf{u} \in N_\mathbf{x}(\mathcal{M})$ be the normal direction at $\mathbf{x}$ with $\|\mathbf{u}\|_2 = 1$. Since $r < \Delta$, we can find $r_0 > r$ such that $r_0 < \Delta$ and $r_0 < \delta$. Denote

$$\mathbf{x}_a := \mathbf{x} + r_0 \mathbf{u}. \tag{B.40}$$

Clearly, we have $\mathbf{x}_a \in B(\mathbf{x}, \delta)$. Since $\mathcal{N}_\Delta(\mathcal{M})$ is a tubular neighborhood of $\mathcal{M}$, we have

$$d_2(\mathbf{x}_a, \mathcal{M}) = r_0 > r, \tag{B.41}$$

which implies that $\mathbf{x}_a$ is an off-manifold adversarial example of $f_t$ at $\mathbf{x}$. Notice that Equation (B.41) not necessarily holds when $r_0 > \Delta$. □

**Corollary 5.3.** *Let $f_b$ be a semantic classifier with semantic information $A_f$ and $B_f$ that satisfy a $2\lambda$-separated property. Given $\epsilon > 0$, there is a ReLU network $\tilde{f}$ with $O((1/\lambda\epsilon)^d) \cdot O(d^2 + d\log(1/\epsilon))$ parameters such that $\|f - \tilde{f}\|_\infty \le \epsilon$.*

*Proof of Corollary 5.3.* According to Lemma 5.2, our goal is to upper bound the Lipschitz constant $l$ of $f_b$. By definition, it suffices to upper bound the supremum of

$$
\begin{aligned}
s := & \frac{|f_b(\mathbf{x}_1; A_f, B_f) - f_b(\mathbf{x}_2; A_f, B_f)|}{d_p(\mathbf{x}_1, \mathbf{x}_2)} \\
= & \frac{1}{d_p(\mathbf{x}_1, \mathbf{x}_2)} \cdot \left| \frac{d_p(\mathbf{x}_1, A_f)}{d_p(\mathbf{x}_1, A_f) + d_p(\mathbf{x}_1, B_f)} - \frac{d_p(\mathbf{x}_2, A_f)}{d_p(\mathbf{x}_2, A_f) + d_p(\mathbf{x}_2, B_f)} \right|.
\end{aligned}
\tag{B.42}
$$

We only need to consider three cases:

1. both of $\mathbf{x}_1, \mathbf{x}_2 \in A_f \cup B_f$, or

2. both of $\mathbf{x}_1, \mathbf{x}_2 \in (A_f \cup B_f)^c$, and

3. either $\mathbf{x}_1$ or $\mathbf{x}_2$ is in $A_f \cup B_f$.

When $\mathbf{x}_1, \mathbf{x}_2 \in A_f \cup B_f$, a trivial verification shows that $s \leq \frac{1}{\lambda}$. We now turn to the second case. By symmetry, let

$$
\frac{d_p(\mathbf{x}_1, A_f)}{d_p(\mathbf{x}_1, A_f) + d_p(\mathbf{x}_1, B_f)} - \frac{d_p(\mathbf{x}_2, A_f)}{d_p(\mathbf{x}_2, A_f) + d_p(\mathbf{x}_2, B_f)} > 0.
\tag{B.43}
$$

By simplifying Equation (B.42), we can obtain that

$$
\begin{aligned}
& \frac{|f_b(\mathbf{x}_1; A_f, B_f) - f_b(\mathbf{x}_2; A_f, B_f)|}{d_p(\mathbf{x}_1, \mathbf{x}_2)} \\
= & \frac{1}{d_p(\mathbf{x}_1, \mathbf{x}_2)} \cdot \left( \frac{d_p(\mathbf{x}_1, A_f)}{d_p(\mathbf{x}_1, A_f) + d_p(\mathbf{x}_1, B_f)} - \frac{d_p(\mathbf{x}_2, A_f)}{d_p(\mathbf{x}_2, A_f) + d_p(\mathbf{x}_2, B_f)} \right) \\
\leq & \frac{1}{2\lambda} \cdot \left( \frac{d_p(\mathbf{x}_1, A_f) - d_p(\mathbf{x}_2, A_f)}{d_p(\mathbf{x}_1, \mathbf{x}_2)} \cdot \frac{d_p(\mathbf{x}_2, B_f)}{d_p(\mathbf{x}_2, A_f) + d_p(\mathbf{x}_2, B_f)} \right. \\
& \left. + \frac{d_p(\mathbf{x}_1, B_f) - d_p(\mathbf{x}_2, B_f)}{d_p(\mathbf{x}_1, \mathbf{x}_2)} \cdot \frac{d_p(\mathbf{x}_2, A_f)}{d_p(\mathbf{x}_2, A_f) + d_p(\mathbf{x}_2, B_f)} \right) \\
\leq & \frac{1}{2\lambda} \cdot (1 \cdot 1 + 1 \cdot 1) = \frac{1}{\lambda},
\end{aligned}
\tag{B.44}
$$

which implies that $s \leq \frac{1}{\lambda}$ in this case. Finally, we consider the third case. We assume WLOG that $\mathbf{x}_1 \in A_f$ and $\mathbf{x}_2 \in (A_f \cup B_f)^c$. Substitute into Equation (B.42), we have

$$
\begin{aligned}
s = & \frac{1}{d_p(\mathbf{x}_1, \mathbf{x}_2)} \cdot \left| \frac{d_p(\mathbf{x}_1, A_f)}{d_p(\mathbf{x}_1, A_f) + d_p(\mathbf{x}_1, B_f)} - \frac{d_p(\mathbf{x}_2, A_f)}{d_p(\mathbf{x}_2, A_f) + d_p(\mathbf{x}_2, B_f)} \right| \\
= & \frac{1}{d_p(\mathbf{x}_1, \mathbf{x}_2)} \cdot \frac{d_p(\mathbf{x}_2, A_f)}{d_p(\mathbf{x}_2, A_f) + d_p(\mathbf{x}_2, B_f)} \leq \frac{1}{2\lambda}
\end{aligned}
\tag{B.45}
$$

To sum up above, we have $\sup_{\mathbf{x}_1 \neq \mathbf{x}_2} s = \frac{1}{\lambda}$, which implies that $f_b$ is $\frac{1}{\lambda}$-Lipschitz continuous, as is required. □

**Corollary 5.3.** *Given $\epsilon > 0$, $r > 0$ and $S \subset [0, 1]^d$, there is a ReLU network $\tilde{\phi}$ with $O((1/r\epsilon)^d) \cdot O(d^2 + d \log(1/\epsilon))$ parameters that can approximate $\phi(\cdot; r, S)$ to precision $\epsilon$.*

*Proof of Corollary 5.3.* We prove this corollary in a similar manner as Corollary 5.3, i.e., we upper bound the supremum of

$$
\begin{aligned}
s := & \frac{|\phi(\mathbf{x}_1; r, S) - \phi(\mathbf{x}_2; r, S)|}{d_p(\mathbf{x}_1, \mathbf{x}_2)} = \frac{1}{d_p(\mathbf{x}_1, \mathbf{x}_2)} \cdot \left| \frac{d_p(\mathbf{x}_1, S)}{r + d_p(\mathbf{x}_1, S)} - \frac{d_p(\mathbf{x}_2, S)}{r + d_p(\mathbf{x}_2, S)} \right| \\
= & \frac{r}{d_p(\mathbf{x}_1, \mathbf{x}_2)} \cdot \left| \frac{1}{r + d_p(\mathbf{x}_1, S)} - \frac{1}{r + d_p(\mathbf{x}_2, S)} \right|.
\end{aligned}
\tag{B.46}
$$

We also consider three cases in this proof:

1. both of $\mathbf{x}_1, \mathbf{x}_2 \in S$, or

2. both of $\mathbf{x}_1, \mathbf{x}_2 \in S^c$, and

3. either $\mathbf{x}_1$ or $\mathbf{x}_2$ is in $S$.

In case 1, we see at once that $s = 0$. When both of $\mathbf{x}_1, \mathbf{x}_2 \in S^c$, we assume WLOG that $d_p(\mathbf{x}_1, S) > d_p(\mathbf{x}_2, S)$. By Equation (B.46), we have

$$s = \frac{d_p(\mathbf{x}_1, S) - d_p(\mathbf{x}_2, S)}{d_p(\mathbf{x}_1, \mathbf{x}_2)} \cdot \frac{r}{(r + d_p(\mathbf{x}_1, S))(r + d_p(\mathbf{x}_2, S))} \leq \frac{1}{r}. \tag{B.47}$$

Analysis similar to Equation (B.47) shows that

$$s = \frac{r}{d_p(\mathbf{x}_1, \mathbf{x}_2)} \cdot \left( \frac{1}{r} - \frac{1}{r + d_p(\mathbf{x}_2, S)} \right) \leq \frac{1}{r}. \tag{B.48}$$

To sum up above, we have $\sup_{\mathbf{x}_1 \neq \mathbf{x}_2} s = \frac{1}{\lambda}$, which implies that $\phi$ is $\frac{1}{r}$-Lipschitz continuous, as is required. $\qquad \square$

**Proposition 5.6.** *Given $\epsilon, \lambda, \delta, r > 0$, for any $f \in \mathcal{F}_{\mathcal{M}}$, there is a ReLU network $\tilde{f}$ with*

$$O(\max\{\frac{1}{\lambda\epsilon}, \frac{2}{r\epsilon}\}^d) \cdot O(d^2 + d\log(\frac{1}{\epsilon})) + O(\log^2(\frac{1}{\epsilon})) \tag{B.49}$$

*parameters that satisfies $\|f - \tilde{f}\|_\infty \leq \epsilon$.*

*Proof of Proposition 5.6.* This proposition can be derived directly from Lemma 5.5, Corollary 5.3, and Corollary 5.4. $\qquad \square$

**Theorem 5.7.** *Consider TBAs with perturbation radius $\delta \in (0, \lambda/2]$, target model $f_t = f_b^* \cdot \phi_{\text{off}}$ and source model $f_s = f_b \cdot \phi_{\text{off}}$, $f_b \in \mathcal{F}_b$. Denote the semantic information of $f_b$ by $A_f$ and $B_f$. Given $\epsilon \leq 0.1$, let $\tilde{f}_t$ and $\tilde{f}_s$ be ReLU networks that satisfy*

$$\|\tilde{f}_t - f_t\|_\infty \leq \epsilon, \|\tilde{f}_s - f_s\|_\infty \leq \epsilon \tag{B.50}$$

*Then, for $\forall \mathbf{x} \in (A \cup B) \cap (A_f \cup B_f)$, the adversarial examples $\mathbf{x}_a$ (if exist) of $\tilde{f}_s$ satisfies*

$$\tilde{f}_t(\mathbf{x}) \cdot \tilde{f}_t(\mathbf{x}_a) \leq 2\epsilon(1 + \epsilon)^2 + 2\epsilon^2. \tag{B.51}$$

*Proof of Theorem 5.7.* Consider $\mathbf{x} \in A_f$ WLOG. If $f_s$ is robust against adversarial examples at $\mathbf{x} \in B(\mathbf{x}, \delta) \cap \mathcal{M} \subset A_f$, then there is nothing to prove. If not, denote the adversarial example of $f_s$ at $\mathbf{x}$ by $\mathbf{x}_a$. Since $\mathbf{x} \in A_f \subset \mathcal{M}$, there is

$$\tilde{f}_s(\mathbf{x}) \geq \tilde{f}_b(\mathbf{x}) \cdot \tilde{\phi}_{\text{off}}(\mathbf{x}) - \epsilon \geq (1 - \epsilon)^2 - \epsilon > 0 \tag{B.52}$$

and we thus have $\tilde{f}_s(\mathbf{x}_a) < 0$. By $\mathbf{x}_a \in B(\mathbf{x}; \delta)$ and the assumption $\delta < \frac{\lambda}{2}$, we have

$$\tilde{f}_b(\mathbf{x}_a) = 1 - \frac{2d_p(\mathbf{x}_a, A_f)}{d_p(\mathbf{x}_a, B_f) + d_p(\mathbf{x}_a, A_f)} \geq 1 - \frac{2\delta}{2\lambda} \geq \frac{1}{2}. \tag{B.53}$$

To obtain $\tilde{\times}(\tilde{f}_b, \tilde{\phi}_{\text{off}})(\mathbf{x}_a) < 0$, there must be $\tilde{f}_b(\mathbf{x}_a) \cdot \tilde{\phi}_{\text{off}}(\mathbf{x}_a) < \epsilon$, which implies that

$$\tilde{\phi}_{\text{off}}(\mathbf{x}_a) < 2\epsilon. \tag{B.54}$$

Now consider $\tilde{f}_t(\mathbf{x})$ and $\tilde{f}_t(\mathbf{x}_a)$. By definition, we have $\tilde{f}_b^*(\mathbf{x}) \in [1 - \epsilon, 1 + \epsilon]$, $\tilde{\phi}_{\text{off}}(\mathbf{x}) \in [1 - \epsilon, 1 + \epsilon]$, and thus

$$\tilde{f}_t(\mathbf{x}) = \tilde{\times}(\tilde{f}_b^*, \tilde{\phi}_{\text{off}})(\mathbf{x}) \leq (1 + \epsilon)^2 + \epsilon. \tag{B.55}$$

Similar to Equation (B.53), there is

$$\tilde{f}_b^*(\mathbf{x}_a) = 1 - \frac{2d_p(\mathbf{x}_a, A)}{d_p(\mathbf{x}_a, B) + d_p(\mathbf{x}_a, A)} \leq 1 \tag{B.56}$$

Combining Equations (B.54) to (B.56) together, we have

$$\tilde{f}_t(\mathbf{x}) \cdot \tilde{f}_t(\mathbf{x}_a) \leq 2\epsilon(1 + \epsilon)^2 + 2\epsilon^2. \tag{B.57}$$

as is required. $\qquad \square$

**Theorem 5.8.** *Consider TBAs with perturbation radius $\delta \in (0, \lambda/2]$, target model $f_t = f_b^* \cdot \phi_{\text{off}}$ and source model $f_s = f_b \cdot \phi_{\text{off}}$, $f_b \in \mathcal{F}_b$. Denote the semantic information of $f_b$ by $A_f$ and $B_f$. Given $\epsilon \le 0.1$, let $\tilde{f}_t$ and $\tilde{f}_s$ be ReLU networks that satisfy Equation (13). Then, for $\forall \mathbf{x} \in A \cup B$, we have*

1. *if $B(\mathbf{x}, \delta) \cap \mathcal{M} \subset S_{\text{crt}} \cup S_{\text{wrg}}$, then $\tilde{f}_t$ and $\tilde{f}_s$ are both robust against adversarial examples;*

2. *if $B(\mathbf{x}, \delta) \cap \mathcal{M} \subset A \cup B$ and $\mathbf{x} \in A_f \cup B_f$, then the adversarial examples of $\tilde{f}_s$ at $\mathbf{x}$ (if exists) cannot transfer to $\tilde{f}_t$.*

*Proof of Theorem 5.8.* The proof of the first result is also straightforward, which is omitted here. It remains to prove the second result. By Equation (7), denote

$$f_s(\mathbf{x}) = f_b(\mathbf{x}) \cdot \phi_{\text{on}}(\mathbf{x}; f_b) = \frac{d_2(\mathbf{x}, B_f) - d_2(\mathbf{x}, A_f)}{d_2(\mathbf{x}, B_f) + d_2(\mathbf{x}, A_f)} \cdot \frac{\alpha\delta - d_2(\mathbf{x}, \mathcal{N}_\delta(A_f \cup B_f))}{\alpha\delta + d_2(\mathbf{x}, \mathcal{N}_\delta(A_f \cup B_f))}. \quad (\text{B.58})$$

For $\forall \mathbf{x} \in A \cup B$ such that $B(\mathbf{x}, \delta) \cap \mathcal{M} \subset A \cup B$ and $\mathbf{x} \in A_f \cup B_f$, denote the unspecific adversarial example (if exist) of $f_s$ at $\mathbf{x}$ by $\mathbf{x}_a$. Assume that $\mathbf{x} \in A \cap A_f$ WLOG. By definition, we have

$$\tilde{f}_b(\mathbf{x}) \in [1 - \epsilon, 1 + \epsilon]. \quad (\text{B.59})$$

By the $2\lambda$-separated assumption of $A_f$ and $B_f$, we have and

$$\tilde{f}_b(\mathbf{x}_a) \in [1 - \epsilon, 1 + \epsilon]. \quad (\text{B.60})$$

Since $\mathbf{x} \in A_f \cup B_f$, we have $\mathbf{x} \in \mathcal{N}_\delta(A_f \cup B_f)$ and

$$\tilde{\phi}_{\text{on}}(\mathbf{x}; f_b) \in [1 - \epsilon, 1 + \epsilon], \quad (\text{B.61})$$

which implies that

$$\tilde{f}_s(\mathbf{x}) = \tilde{\times}(\tilde{f}_b, \tilde{\phi}_{\text{on}}(\cdot; f_b))(\mathbf{x}) \ge (1 - \epsilon)^2 - \epsilon > 0. \quad (\text{B.62})$$

From $\tilde{f}_s(\mathbf{x})\tilde{f}_s(\mathbf{x}_a) < 0$, we can obtain that $\tilde{f}_s(\mathbf{x}_a) < 0$, which implies that

$$\tilde{\phi}_{\text{on}}(\mathbf{x}_a; f_b) \cdot \tilde{f}_b(\mathbf{x}) < \epsilon, \quad (\text{B.63})$$

which implies that

$$\tilde{\phi}_{\text{on}}(\mathbf{x}_a; f_b) < \frac{\epsilon}{1 - \epsilon} < 2\epsilon. \quad (\text{B.64})$$

By definition, we have

$$\tilde{f}_t(\mathbf{x}) = \tilde{\times}(\tilde{f}_b^*, \tilde{\phi}_{\text{on}}(\cdot; f_b^*))(\mathbf{x}) \ge (1 - \epsilon)^2 - \epsilon > 0. \quad (\text{B.65})$$

By $\mathbf{x} \in A$ and the $2\lambda$-separated assumption of $A$ and $B$, we have

$$\tilde{f}_b^*(\mathbf{x}_a) = 1 - \frac{2d_p(\mathbf{x}_a, A)}{d_p(\mathbf{x}_a, B) + d_p(\mathbf{x}_a, A)} \ge 1 - \frac{2\delta}{2\lambda} \ge \frac{1}{2}. \quad (\text{B.66})$$

By $B(\mathbf{x}, \delta) \cap \mathcal{M} \subset A$, we have $\mathbf{x}_a \in \mathcal{N}_\delta(A \cup B))$, i.e.,

$$\phi_{\text{on}}(\mathbf{x}_a; f_b^*) \in [1 - \epsilon, 1 + \epsilon] \quad (\text{B.67})$$

Together, we have

$$\tilde{f}_t(\mathbf{x}_a) = \tilde{\times}(\tilde{f}_b^*, \tilde{\phi}_{\text{on}}(\cdot; f_b^*))(\mathbf{x}_a) \ge \frac{1 - \epsilon}{2} > 0, \quad (\text{B.68})$$

which completes the proof. □

**Proposition 5.9.** *For any classifier $f^*$ with $R_{\text{std}}(f^*) = 0$ and perturbation radius $\delta \in (r_\delta(f^*), \lambda)$, there is $f \in \mathcal{F}_{\mathcal{M}}$ such that*

1. *$R_{\text{std}}(f) = R_{\text{std}}(f^*)$, and*

2. *for $\forall \mathbf{x} \in A \cup B$, if $\mathbf{x}_a$ is an adversarial example of $f^*$ at $\mathbf{x}$, then exists $\mathbf{x}_a' \in B(\mathbf{x}_a, r_\delta(f^*)/4)$ such that $\mathbf{x}_a'$ is an adversarial example of $f$.*

*Proof of Proposition 5.9.* Define the following set

$$S_a := \{\mathbf{x} \in [0,1]^d \cap (A \cup B)^c : \exists \mathbf{x}' \in A \cup B \ s.t. \ \mathbf{x}' \in B(\mathbf{x}; \delta), f^*(\mathbf{x})f^*(\mathbf{x}') < 0\}, \tag{B.69}$$

and let

$$G = \left( \bigcup_{\mathbf{x} \in S_a} B(\mathbf{x}, r_\delta(f^*)/2) \right)^c. \tag{B.70}$$

By definition, $A \cup B \in G$. Consider

$$f(\mathbf{x}) = f_b^*(\mathbf{x}) \cdot \phi(\mathbf{x}; r_\delta(f^*)/4, G). \tag{B.71}$$

Since $A \cup B \in G$, we have $R_{\text{std}}(f) = R_{\text{std}}(f_b^*) = 0 = R_{\text{std}}(f)^*$. For $\forall \mathbf{x} \in A \cup B$, if $\mathbf{x}_a$ is an adversarial example of $f^*$ at $\mathbf{x}$, then

$$\frac{r_\delta(f^*)}{4} \le d_p(\mathbf{x}_a, G) \le \frac{r_\delta(f^*)}{2}, \tag{B.72}$$

which implies that exists $\mathbf{x}'_a \in B(\mathbf{x}_a, r_\delta(f^*)/4)$ such that $\mathbf{x}'_a$ is an adversarial example of $f$. $\qquad \square$

