# A Theory of Transfer-Based Black-Box Attacks: Explanation and Implications (Supplementary Material)

This article serves as the supplementary material to the central part of our paper. Appendix A includes some further discussions. Complete proofs of the theorems and propositions in Sections 4 and 5 can be found in Appendix B. A multi-class analysis of the manifold attack model is given in Appendix C.

## A  Further Discussions

### A.1  What makes a good explanatory model?

As its title suggests, our paper's primary effort is to explain the properties of TBAs by the manifold attack model. During the writing of this paper, the following question is discussed repeatedly:

*What makes a good explanatory model and how to evaluate an explanatory model?*

This subsection provides our answer to this question. First of all, we believe that a good explanatory model should be:

- **(Criterion 1)** consistent with existing empirical results,

- **(Criterion 2)** based on reasonable assumptions, and

- **(Criterion 3)** theoretically tractable.

Throughout this paper, we make many efforts to validate our model. Specifically, we try to check whether our model fulfills criteria 1-3. Clearly, our model is theoretically tractable. We theoretically analyze TBAs and provide many explanatory results in Sections 4 and 5.

In the rest of this subsection, we briefly discuss criteria 1 and 2. For the first criterion, we discuss the intriguing properties of TBAs (i.e., the empirical results observed by previous works) in Sections 1 and 2. Two of the most widely-known properties of TBAs are: 1) TBAs can craft transferable adversarial examples even when the source model is inaccurate [1], and 2) the success rates of TBAs are constantly lower than other methods of black-box adversarial attacks [2–4]. Section 4 demonstrates that our model is consistent with the existing empirical results and provides reasonable explanations for these properties.

As for criterion 2, our model assumes that the natural data lies on a low-dimensional manifold. This assumption is commonly seen in previous works [5–8]. We also assume that the classifiers (i.e., the source and target models in TBAs) can be decomposed into the product of a semantic classifier $f_b$ (Definition 4.1) and a concentration multiplier $\phi$ (Definition 4.2). This assumption is based on the empirical observation that *ML models can capture semantic and geometrical information of the natural data* [9, 10]. Here, our concerns are two folds: 1) what are the semantic and geometrical information, and 2) how does an ML model capture such information?

Submitted to 37th Conference on Neural Information Processing Systems (NeurIPS 2023). Do not distribute.

**Semantic information**  We first focus on semantic information. The following remark explains what is the semantic information of a dataset by an example.

*Remark* A.1 (The semantic information of CIFAR-10). Generally speaking, "semantic" refers to the relationship between natural data and their true label, which should be consistent with human recognition. For example, the semantic information contained in the CIFAR-10 dataset is the true labels (e.g., airplane, automobile, and bird) and their corresponding natural images (e.g., images of airliners, SUVs, and chickens). In this example, an image cannot simultaneously include an airplane and an automobile since "the classes are completely mutually exclusive" in the CIFAR-10 dataset, cf. the official website of CIFAR-10. That is, the semantic information provided by CIFAR-10 is separated.  ▲

In our paper, we formalize the semantic information of natural data by separated sets $A^1, A^2, \cdots, A^k \subset \mathcal{M}$ (for a $k$-class classification task), see Section 3.2 for the definitions. As is discussed in Remark A.1, these sets reflect the relationship between true labels and their corresponding natural data, and more importantly, these sets should be separated. In this paper, we define separated sets in Definition 3.2 and assume that $A^1, A^2, \cdots, A^k \subset \mathcal{M}$ are separated. It is worth noting that the definition of "semantic information" in our paper is motivated by that of the "concept" in classical learning theory [11, 12]. In these works, learning a concept is equivalent to approximating the decision boundary of ML models to the concept sets (i.e., subsets in the sample space).

Our model captures the semantic information in a similar way as [11]. We let $A_f^1, A_f^2, \cdots, A_f^k \subset \mathcal{M}$ be the semantic information learned by $f$. Note that we do not assume these sets to be regions or to have any compactness or connectedness restriction. Instead, we only assume that these sets are separated (as the semantic information of natural data). The "similarity" between $A_f^i$ and $A^i$ ($1 \leq i \leq k$) reflects how well the ML model $f$ has learned the semantic information of the training data.

**Geometrical information**  As for the geometrical information, we are motivated by the methods in OOD detection [13–16]. In these works, the scores of the OOD samples are lower than in-distribution samples. In our setting, by the low-dimensional manifold assumption, we know that the off-manifold data are also outside of the data distribution. Thus, by approximating the shape of the manifold, the concentration multiplier $\phi$ should assign lower scores to those off-manifold samples, see Definition 4.2 for a formal definition. In summary, our paper assumes that natural data is drawn from a low-dimensional manifold and the source and target models capture the semantic and geometrical information in the way we have discussed above. Our assumption is intuitive, reasonable, and milder than previous works that theoretically analyze TBAs. Our model fulfills criterion 2.

Last but not least, the following remark explains why our paper does not present any experiments.

*Remark* A.2 (Experiments are unnecessary for validating our model). As mentioned in Section 2.1, most of the recent studies on TBAs focus on empirically improving the success rates of TBAs [3, 17]. However, to the best of our knowledge, existing theoretical analyses of TBAs [18–20] are either based on simple models (e.g., linear classifiers) or strong assumptions (e.g., natural data are drawn from the spherical Gaussian distribution). The theoretical studies of TBAs are falling behind the engineering practice, which motivates us to propose an explanatory model that analyzes and explains the existing empirical results. As is discussed in Appendix A.1, we argue that conducting experiments (on either real-world or synthetic datasets) is unnecessary for evaluating an explanatory model. Therefore, we do not include experiments in our paper.  ▲

### A.2  Visualization of the Non-Adversarial Region

We provide a visualization of Example 4.10 in Figure A.1.

## B  Complete Proofs

**Proposition 4.3** (semantic classifier, binary case). *Given $2\lambda$-separated sets $A_f, B_f \subset \mathcal{M}$. Define:*

$$f_b(\mathbf{x}) = f_b(\mathbf{x}; A_f, B_f) := \frac{d_p(\mathbf{x}, B_f) - d_p(\mathbf{x}, A_f)}{d_p(\mathbf{x}, B_f) + d_p(\mathbf{x}, A_f)}. \tag{B.1}$$

*Then, $f_b$ is a semantic classifier. In particular, we can obtain from Equation (B.1) that $f_b(\mathbf{x}) > 0$ if $\mathbf{x}$ is closer (w.r.t. $d_p$) to $A_f$ than $B_f$ and $f_b(\mathbf{x}) < 0$ otherwise.*

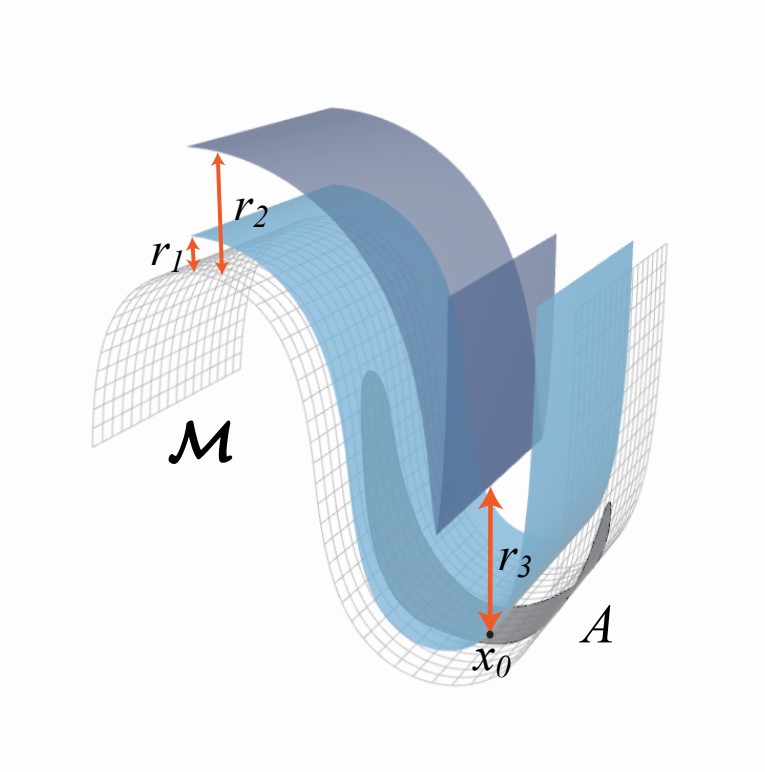

Figure A.1: A visualization of Example 4.10. The data manifold $\mathcal{M}$ is represented by the grid surface. Let the surface in light blue (or dark blue) be the contour surface that $\phi_1 = 0$ (or $\phi_2 = 0$). The distance between $\mathbf{x}_0$ and the dark blue surface is $r_3$, which is greater than $\delta$ and $r_2$.

*Proof of Proposition 4.3.* It is easy to check that $f_b(\mathbf{x}) = 1$ when $\mathbf{x} \in A_f$ and $f_b(\mathbf{x}) = -1$ when $\mathbf{x} \in B_f$. By definition, we know that $f_b$ is a semantic classifier. $\square$

**Proposition 4.4.** *Take $A_f = A$ and $B_f = B$ in Equation* (B.1) *and denote the corresponding classifier by $f_b^*$. Then, for any given $\lambda \geq \delta > 0$, we have $R_{\mathrm{std}}(f_b^*) = R_{\mathrm{adv}}(f_b^*, \delta) = 0$.*

*Proof of Proposition 4.4.* By Equation (B.1), we have

$$f_b^*(\mathbf{x}) = \frac{d_p(\mathbf{x}, B) - d_p(\mathbf{x}, A)}{d_p(\mathbf{x}, B) + d_p(\mathbf{x}, A)}, \ \forall \mathbf{x} \in \mathbb{R}^d. \tag{B.2}$$

Clearly, we have $f_b^*(\mathbf{x}) = 1$ when $\mathbf{x} \in A$ and $f_b^*(\mathbf{x}) = -1$ when $\mathbf{x} \in B$. Then, the standard risk of $f_b^*$ w.r.t. $D(\mathbf{x})$ is

$$R_{\mathrm{std}}(f_b^*) = \mathbb{P}_D\left[f_b^*(\mathbf{x})y < 0 \mid \mathbf{x} \in A\right] + \mathbb{P}_D\left[f_b^*(\mathbf{x})y < 0 \mid x \in B\right] = 0 \tag{B.3}$$

Recall that $A$ and $B$ are $2\lambda$-separated (cf. Definition 3.2). For $\forall \mathbf{x} \in A$ and $\mathbf{x}' \in B(\mathbf{x}, \delta)$, we have $d_p(\mathbf{x}, B) > \delta$, which implies that $d_p(\mathbf{x}', B) - d_p(\mathbf{x}', A) > 0$, and thus $f_b^*(\mathbf{x}')f_b^*(\mathbf{x}) = f_b^*(\mathbf{x}') > 0$. For $\forall \mathbf{x} \in B$, a similar deduction shows that $f_b^*(\mathbf{x}')f_b^*(\mathbf{x}) > 0$ holds for $\forall \mathbf{x}' \in B(\mathbf{x}, \delta)$. Together, we have

$$\begin{aligned} R_{\mathrm{adv}}(f_b^*, \delta) := &\mathbb{P}\left[\exists \mathbf{x}' \in B(\mathbf{x}; \delta) \ s.t. \ f_b^*(\mathbf{x}')f_b^*(\mathbf{x}) < 0 \mid \mathbf{x} \in A\right] \\ &+ \mathbb{P}\left[\exists \mathbf{x}' \in B(\mathbf{x}; \delta) \ s.t. \ f_b^*(\mathbf{x}')f_b^*(\mathbf{x}) < 0 \mid \mathbf{x} \in B\right] = 0, \end{aligned} \tag{B.4}$$

which completes the proof. $\square$

*Remark* B.1. The construction of Equation (B.2) can be found in previous works [6, 21]. In particular, Li et al. [6] uses the ReLU-approximation of $f_b^*$ to study the robust generalization of deep NNs.

**Proposition 4.5** (Concentration multiplier, binary case). *For any given $r > 0$ and $G \subset \mathbb{R}^d$, denote*

$$\phi(\mathbf{x}) = \phi(\mathbf{x}; r, G) := \frac{r - d_p(\mathbf{x}, G)}{r + d_p(\mathbf{x}, G)}, \ \forall \mathbf{x} \in \mathbb{R}^d. \tag{B.5}$$

*Then $\phi(\mathbf{x})$ is a concentration multiplier around $G$.*

*Proof of Proposition 4.5.* For $\forall \mathbf{x} \in G$, we have $d_p(\mathbf{x}, G) = 0$. That is, $\phi(\mathbf{x}) = 1$ for $\forall \mathbf{x} \in G$. For $\forall \mathbf{x}_1, \mathbf{x}_2$ s.t. $d_p(\mathbf{x}_1, G) > d_p(\mathbf{x}_2, G)$, it is easy to check that $\phi(\mathbf{x}_1) < \phi(\mathbf{x}_2)$. □

**Proposition 4.6.** *Let $f = f_b \cdot \phi$ and $A_f, B_f$ be the semantic information of $f_b$. We can obtain that*

1. *if $R_{\mathrm{adv}}(f; \delta) \neq 0$, then $f$ suffers from off-manifold adversarial examples.*

2. *if $R_{\mathrm{adv}}(f; \delta) \neq 0$ and $d_p(A \cup B, (A_f \cup B_f)^c) > \delta$, then all the adversarial examples of $f$ are off the manifold.*[1]

*Proof of Proposition 4.6.* We first prove the first result. By definition, there are $r > 0$ and $G \subset \mathbb{R}^d$ such that

$$f(\mathbf{x}) = \frac{d_p(\mathbf{x}, B_f) - d_p(\mathbf{x}, A_f)}{d_p(\mathbf{x}, B_f) + d_p(\mathbf{x}, A_f)} \cdot \frac{r - d_p(\mathbf{x}, G)}{r + d_p(\mathbf{x}, G)} \tag{B.6}$$

Given $R_{\mathrm{adv}}(f; \delta) \neq 0$, then $\exists \mathbf{x} \in A \cup B$ and $\mathbf{x}_0 \in B(\mathbf{x}, \delta)$ such that $f(\mathbf{x}) f(\mathbf{x}_0) < 0$. If $x_0 \in \mathcal{M}^c$, there is nothing to prove.

Otherwise, we have $\mathbf{x}_0 \in \mathcal{M}$. Without loss of generality (WLOG), we assume that such $\mathbf{x} \in A$ and $f(\mathbf{x}_0) < 0$, which implies that either $f_b(\mathbf{x}_0) < 0$ or $\phi(\mathbf{x}_0) < 0$. We first consider the case when $f_b(\mathbf{x}_0) < 0$. Then, we have $d_p(\mathbf{x}, B_f) - d_p(\mathbf{x}, A_f) < 0$ and $r - d_p(\mathbf{x}, S) > 0$. Denote

$$r_0 := \frac{1}{3} \min\{|d_p(\mathbf{x}_0, A_f) - d_p(\mathbf{x}_0, B_f)|, |r - d_p(\mathbf{x}_0, S)|, \delta\}. \tag{B.7}$$

Consider the non-empty set

$$B(\mathbf{x}, \delta) \cap B(\mathbf{x}_0, r_0) \cap \mathcal{M}^c.$$

For $\forall x_0' \in B(\mathbf{x}_0, r_0)$, there is

$$d_p(\mathbf{x}_0', A_f) \geq d_p(\mathbf{x}_0, A_f) - d_p(\mathbf{x}_0, \mathbf{x}_0'), \tag{B.8}$$

and

$$d_p(\mathbf{x}_0', B_f) \leq d_p(\mathbf{x}_0, B_f) + d_p(\mathbf{x}_0, \mathbf{x}_0'), \tag{B.9}$$

which implies that

$$d_p(\mathbf{x}_0', A_f) - d_p(\mathbf{x}_0', B_f) \geq d_p(\mathbf{x}_0, A_f) - d_p(\mathbf{x}_0, B_f) - 2 d_p(\mathbf{x}_0, \mathbf{x}_0') \geq r_0 > 0. \tag{B.10}$$

Similarly, we can obtain $r - d_p(\mathbf{x}_0', S) > 0$. Together, these two inequalities lead us to $f(\mathbf{x}_0') = f(\mathbf{x}_0')f(\mathbf{x}) < 0$, i.e., $x_0'$ is an off-manifold adversarial example of $x_0$. Some tedious manipulation yields the same result when $\phi(\mathbf{x}_0) < 0$, which is omitted here.

As for the second result, since $R_{\mathrm{adv}}(f; \delta) \neq 0$, we can obtain from the first result that off-manifold adversarial examples exist. It remains to show that $f$ has no on-manifold adversarial examples.

Since $d_p(A \cup B, (A_f \cup B_f)^c) > \delta$ and by assumption $A_f \cup B_f \subset G$, we have

$$\frac{r - d_p(\mathbf{x}, G)}{r + d_p(\mathbf{x}, G)} = 1 \tag{B.11}$$

and

$$\frac{r - d_p(\mathbf{x}', G)}{r + d_p(\mathbf{x}', G)} = 1 \tag{B.12}$$

for $\forall \mathbf{x} \in A \cup B$ and $\mathbf{x}' \in B(\mathbf{x}, \delta) \cap \mathcal{M}$. We can easily obtain that $f_b(\mathbf{x}) = f_b(\mathbf{x}')$, which implies that $f$ has no on-manifold adversarial examples. □

---

[1]**Notice:** In the main part of the paper, we made a typo in this result. Here, we provide the corrected version. The other results in the main paper are based on the corrected version of this result.

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

# C  Analyses in Multi-Class Classification Problems

This section some of the results in Sections 4 and 5 to $k$-class classification problems. We first extend Propositions 4.3 and 4.5 to multi-class classification.

**Proposition C.1** (Semantic classifier, multi-class case)**.** *Given $2\lambda$-separated sets $A_f^1, A_f^2, \cdots, A_f^k \subset \mathcal{M}$. Consider $f_b(\mathbf{x}) = (f_b^{(1)}(\mathbf{x}), f_b^{(2)}(\mathbf{x}), \cdots, f_b^{(k)}(\mathbf{x}))^T$ and define:*

$$f_b^{(i)}(\mathbf{x}) := \frac{\left( \sum_{j \neq i} d_p(\mathbf{x}, A_f^j) \right) - d_p(\mathbf{x}, A_f^i)}{\left( \sum_{j \neq i} d_p(\mathbf{x}, A_f^j) \right) + d_p(\mathbf{x}, A_f^i)} \tag{C.57}$$

*for $\forall 1 \le i \le k$. Then, $f_b$ is a semantic classifier.*

*Proof of Proposition C.1.* By Equation (C.57), we have

$$f_b^{(i)}(\mathbf{x}) = \frac{\left( \sum_{j=1}^k d_p(\mathbf{x}, A_f^j) \right) - d_p(\mathbf{x}, A_f^i)}{\sum_{j=1}^k d_p(\mathbf{x}, A_f^j)} \tag{C.58}$$

for $\forall 1 \le i \le k$. Then, there is

$$y(f_b, \mathbf{x}) = \arg\max_{1 \le i \le k} f_b^{(i)}(\mathbf{x}) = \arg\max_{1 \le i \le k} \left( -d_p(\mathbf{x}, A_f^i) \right) = \arg\min_{1 \le i \le k} d_p(\mathbf{x}, A_f^i). \tag{C.59}$$

Given that $A_f^1, A_f^2, \cdots, A_f^k$ are $2\lambda$-separated, we have

$$0 = d_p(\mathbf{x}, A_f^i) < d_p(\mathbf{x}, A_f^j) \tag{C.60}$$

for $\forall j \neq i$ if $\mathbf{x} \in A_f^j$, which completes the proof. $\qquad\square$

Next, we specify a family of concentration multipliers for multi-class TBAs.

**Proposition C.2** (Concentration multiplier, multi-class case)**.** *For any given $r > 0$ and $G \subset \mathbb{R}^d$, denote*

$$\phi(\mathbf{x}) = \phi(\mathbf{x}; r, G) := \frac{r - d_p(\mathbf{x}, G)}{r + d_p(\mathbf{x}, G)}, \quad \forall \mathbf{x} \in \mathbb{R}^d. \tag{C.61}$$

*Then $\phi(\mathbf{x})$ is a concentration multiplier around $G$.*

Note that Equation (C.61) is identical to Equation (5). The proof of Proposition C.2 is therefore omitted. The following proposition extends Proposition 4.4 to the multi-class case.

**Proposition C.3.** *Take $A^i_f = A^i$ in Equation (C.57) for $\forall 1 \le i \le k$. Denote the corresponding classifier by $f^*_b$. Then, for any given $\lambda \ge \delta > 0$, we have $R_{\mathrm{std}}(f^*_b) = R_{\mathrm{adv}}(f^*_b, \delta) = 0$.*

*Proof of Proposition C.3.* By Equation (C.57), we have

$$f^*_b(\mathbf{x}) = \frac{\left(\sum_{j \ne i} d_p(\mathbf{x}, A^j)\right) - d_p(\mathbf{x}, A^i)}{\left(\sum_{j \ne i} d_p(\mathbf{x}, A^j)\right) + d_p(\mathbf{x}, A^i)} \tag{C.62}$$

Apparently, we have $y(f^*_b, \mathbf{x}) = i$ when $\mathbf{x} \in A^i$ for $\forall 1 \le i \le k$. The standard risk of $f^*_b$ w.r.t. $D$ is

$$R_{\mathrm{std}}(f^*_b) = \sum_{i=1}^k \left( \mathbb{P}_D \left[ y(f^*_b, \mathbf{x}) \ne y(\mathbf{x}) \mid \mathbf{x} \in A^i \right] \right) = 0. \tag{C.63}$$

For $\forall i \ne j$, recall that $A$ and $B$ are $2\lambda$-separated (cf. Definition 3.2). For $\forall \mathbf{x} \in A^i$ and $\mathbf{x}' \in B(\mathbf{x}, \delta)$, we have $d_p(\mathbf{x}', A^j) > \delta$, which implies that $d_p(\mathbf{x}', A_j) > d_p(\mathbf{x}', A^i)$ and

$$\left( \sum_{l=1}^k d_p(\mathbf{x}', A^l) \right) - d_p(\mathbf{x}', A^i) > \left( \sum_{l=1}^k d_p(\mathbf{x}', A^l) \right) - d_p(\mathbf{x}', A^j). \tag{C.64}$$

Since $j$ is arbitrarily chosen, and according to Equation (C.58), we have

$$f^{(i)}_b(\mathbf{x}') > f^{(j)}_b(\mathbf{x}') \tag{C.65}$$

holds for $\forall j \ne i$, i.e., $y(f^*_b, \mathbf{x}) = y(f^*_b, \mathbf{x}')$ for $\forall \mathbf{x}' \in B(\mathbf{x}, \delta)$. Then, the adversarial risk of $f^*_b$ is

$$R_{\mathrm{adv}}(f^*_b, \delta) = \sum_{i=1}^k \left( \mathbb{P}_D \left[ \exists \mathbf{x}_a \in B(\mathbf{x}; \delta) \text{ s.t. } y(f, \mathbf{x}) \ne y(f, \mathbf{x}_a) \mid \mathbf{x} \in A^i \right] \right) = 0, \tag{C.66}$$

which completes the proof. □

Next, we go straight for the two explanatory results. We first note that the non-existence of off-manifold adversarial examples is due to the "sharp curvature" of the data manifold. The analyses in Example 4.10 are regardless of whether the task is binary or multi-class. Here, we extend Proposition 4.12 to multi-class cases. Consider TBAs with perturbation radius $\delta \in (0, \lambda]$ For any unspecified $\alpha \in (0, 1)$, let

$$\phi_{\mathrm{off}}(\mathbf{x}) := \phi(\mathbf{x}; \alpha\delta, \mathcal{M}) = \frac{\alpha\delta - d_p(\mathbf{x}, \mathcal{M})}{\alpha\delta + d_p(\mathbf{x}, \mathcal{M})}. \tag{C.67}$$

Recall that Proposition 4.12 is restricted to $p = 2$. The following proposition provides a sufficient condition for the existence of off-manifold adversarial examples in multi-class classification tasks.

**Proposition C.4.** *Given perturbation radius $\delta \in (0, \lambda]$ and target model $f_t = f^*_b \cdot \phi_{\mathrm{off}}$. Let $\Delta$ be the constant specified in Lemma 4.11. Then, for $\forall \mathbf{x} \in \cup_{i=1}^k A^i$, the off-manifold adversarial example of $f_t$ at $\mathbf{x}$ exists if $\alpha\delta < \Delta$.*

*Proof of Proposition C.4.* For $\forall \mathbf{x} \in \cup_{i=1}^k A^i$, let $\mathbf{u} \in N_{\mathbf{x}}(\mathcal{M})$ be the normal direction at $\mathbf{x}$ with $\|\mathbf{u}\|_2 = 1$. Since $\alpha\delta < \Delta$, we can find $r_0 > \alpha\delta$ such that $r_0 < \Delta$ and $r_0 < \delta$. Denote

$$\mathbf{x}_a := \mathbf{x} + r_0 \mathbf{u}. \tag{C.68}$$

Clearly, we have $\mathbf{x}_a \in B(\mathbf{x}, \delta)$. Since $\mathcal{N}_\Delta(\mathcal{M})$ is a tubular neighborhood of $\mathcal{M}$, we have

$$d_2(\mathbf{x}_a, \mathcal{M}) = r_0 > \alpha\delta. \tag{C.69}$$

For $\forall i \in \{1, 2, \cdots, k\}$, by definition, we have

$$f^{(l)}_t(\mathbf{x}_a) = f^{*,(l)}_b(\mathbf{x}_a) \cdot \phi_{\mathrm{off}}(\mathbf{x}_a) = \frac{\left(\sum_{l \ne i} d_2(\mathbf{x}_a, A^l)\right) - d_2(\mathbf{x}_a, A^i)}{\left(\sum_{l \ne i} d_2(\mathbf{x}_a, A^l)\right) + d_2(\mathbf{x}_a, A^i)} \cdot \frac{\alpha\delta - d_2(\mathbf{x}, \mathcal{M})}{\alpha\delta + d_2(\mathbf{x}, \mathcal{M})}. \tag{C.70}$$

Since $A^1_f, A^2_f, \cdots, A^k_f$ are $2\lambda$-separated, we can easily obtain that

$$y(f^*_b, \mathbf{x}_a) = y(f^*_b, \mathbf{x}). \tag{C.71}$$

Combining Equations (C.69) to (C.71) together, we can obtain that $\mathbf{x}_a$ is an off-manifold adversarial example of $f_t$ at $\mathbf{x}$, since $\phi_{\mathrm{off}}$ is negative and thus turn the arg max of $f^*_b$ to the arg min. □

Let $\mathcal{F}_b$ and $\Phi$ be the function class defined in Proposition C.1 and Proposition C.2, respectively. The following proposition extends Proposition 4.7 to multi-class classification tasks based on the results in Proposition C.4.

**Proposition C.5.** *Denote the target model by $f_t = f_b^* \cdot \phi_{\text{off}}$ and the source model $f_s = f_b \cdot \phi_{\text{off}}$, $f_b \in \mathcal{F}_b$. With some abuse of notation, let the semantic information of $f_b$ be $A_f^1, A_f^2, \cdots, A_f^k$. Let $\Delta$ be the constant specified in Lemma 4.11. Assume that $\alpha\delta < \Delta$. Then, for $\forall \mathbf{x} \in \cup_{i=1}^k A^i$, exists adversarial example of $f_s$ at $\mathbf{x}$ that is transferable if $\mathbf{x} \in \cup_{i=1}^k A_f^i$.*

In the main part of our paper, Proposition 4.7 proves that adversarial examples are transferable even if the source model is accurate, which is consistent with the empirical results in Papernot et al. [1]. Proposition C.5 also explains this phenomenon, even though it is weaker than Proposition 4.7.

*Proof of Proposition C.5.* For $\forall i \in \{1, 2, \cdots, k\}$, we first consider those $\mathbf{x} \in A^i$. By definition, we have

$$f_s^{(l)}(\mathbf{x}) = f_b^{(l)}(\mathbf{x}) \cdot \phi_{\text{off}}(\mathbf{x}) = \frac{\left(\sum_{l \neq i} d_p(\mathbf{x}, A_f^l)\right) - d_p(\mathbf{x}, A_f^i)}{\left(\sum_{l \neq i} d_p(\mathbf{x}, A_f^l)\right) + d_p(\mathbf{x}, A_f^i)} \cdot \frac{\alpha\delta - d_p(\mathbf{x}, \mathcal{M})}{\alpha\delta + d_p(\mathbf{x}, \mathcal{M})}. \tag{C.72}$$

According to Proposition C.4, we know that off-manifold adversarial examples exist. In fact, for $\forall \mathbf{x} \in \cup_{i=1}^k A_f^i$, let $\mathbf{x}_a$ be as defined in Equation (C.68). Similar to the proof of Proposition C.4, we have $\phi_{\text{off}}(\mathbf{x}_a) < 0$ and

$$y(f_b^*, \mathbf{x}_a) = y(f_b^*, \mathbf{x}), \ y(f_b, \mathbf{x}_a) = y(f_b, \mathbf{x}). \tag{C.73}$$

which implies that $\mathbf{x}_a$ is a transferable adversarial example. $\qquad\qquad\square$