# OpenReview forum: "A Theory of Transfer-Based Black-Box Attacks: Explanation and Implications"
_NeurIPS.cc/2023/Conference — NeurIPS 2023 poster_

### Official Review · Reviewer_b7e2 · 2023-06-30

**Soundness:** 3 good
**Presentation:** 3 good
**Contribution:** 4 excellent
**Rating:** 8
**Confidence:** 4

**Summary:**

This paper proposes an explanatory model for transfer-based attacks (TBAs). This model formalizes two popular intuitions and specify a hypothesis class $F_M$. By assuming that the source and target models come from $F_M$, this paper explains the properties of TBA. This paper also discusses some properties of the model itself.

**Strengths:**

+ Technically solid paper with rigorous mathematical proofs.
+ The way that this paper formalizes semantic/geometric information is very interesting. The proposed model greatly simplifies the analysis while not making too much assumptions. This technique might be useful in other fields of research like transfer learning and domain adaptation.

**Weaknesses:**

+ The main results (Section 4) of this paper are not well-organized. It seems that the authors have put all related results in just one section, which would confuse the readers that are not familiar with this topic.

**Questions:**

+ Are there any similar definitions about the semantic/geometrical information in previous works?
+ Whether the manifold attack model can be used to derive probabilistic bound on the success rate of TBAs?

---

> ### Author Rebuttal · Authors · 2023-08-09
>
> Dear Reviewer b7e2:
>
> Thanks to the reviewer for appreciating our effort and providing helpful feedback. We will reply to the reviewer’s concerns point by point in our response.
>
> **(Weakness 1)**
>
> This paper tries to explain several parallel properties of TBAs. As a result, it is hard to integrate the propositions in section 4 into two to three main theorems. As a remedy, we will rewrite the propositions that are directly related to our main explanatory results (i.e., Propositions 4.7 and 4.12) as theorems in future revisions.
>
> **(Question 1)**
>
> In our paper, Definition 4.1 and 4.2 provide the abstract definitions to semantic classifiers and concentration multipliers. As mentioned in section 1 (Lines 45-49), these two concepts are motivated by empirical works, e.g., Ilyas et al., 2019 and Allen-Zhu and Li, 2021. To the best of our knowledge, there is no similar definitions about the semantic classifier or geometrical multiplier in previous works. As for the specific form of $f_b$ and $\phi$ in Proposition 4.3 and 4.5, we are largely inspired by Li et al., 2022, as we have mentioned in the related works.
>
> **(Question 2)**
> Our proposed model can be used to derive probabilistic bound on the success rate of TBAs. In fact, Gilmer et al., 2018 study the transferability of adversarial examples from a probabilistic point of view. The data model in Gilmer et al., 2018 can be view as a special case of our proposed model. However, deriving probabilistic bound require much stronger assumptions on the distribution $D$ and the manifold $\mathcal{M}$, which is beyond the score of our paper.
>
> Thanks again for the reviewer’s comments and feedback. We hope that we have addressed the reviewer's concerns. Please let us know if there are any further questions.
>
> **Reference**
>
> Andrew Ilyas, Shibani Santurkar, Dimitris Tsipras, Logan Engstrom, Brandon Tran, and Aleksander Madry. Adversarial examples are not bugs, they are features. In NeurIPS, 2019.
>
> Zeyuan Allen-Zhu and Yuanzhi Li. Feature purification: How adversarial training performs robust deep learning. In FOCS, 2021.
>
> Binghui Li, Jikai Jin, Han Zhong, John E. Hopcroft, and Liwei Wang. Why robust generalization in deep learning is difficult: Perspective of expressive power. CoRR, abs/2205.13863, 2022.
>
> Justin Gilmer, Luke Metz, Fartash Faghri, Samuel S. Schoenholz, Maithra Raghu, Martin Wattenberg, and Ian J. Goodfellow. Adversarial spheres. In ICLR, 2018.

---

> > ### Comment · Reviewer_b7e2 · 2023-08-15
> > **Final Rate**
> >
> > Thank you for your response, I have no further questions and will maintain my rating.

---

> > > ### Author Response · Authors · 2023-08-15
> > >
> > > Thanks for the response. We sincerely appreciate that.

---

### Official Review · Reviewer_HLto · 2023-07-01

**Soundness:** 3 good
**Presentation:** 3 good
**Contribution:** 3 good
**Rating:** 8
**Confidence:** 4

**Summary:**

The paper focuses on a black-box adversarial attack method called the transfer-based attacks (TBA). The paper establishes a unified framework that explains the properties of TBAs, which has never been done by previous work. More specifically, the paper explains
1. why adversarial examples are transferable when the source model is inaccurate
2. why TBA’s success rate is hard to improve.
The paper also discuss
1. how to approximate their model with ReLU networks
2. the expressive power of their model


**Strengths:**

The paper provides theoretical explanation for the properties of TBA. Comparing to previous works in this area, the paper is based on milder assumptions, provides more delicate results, and can view some previous works as special cases. The paper also discusses the proposed model’s expressive power and approximates the model with ReLU networks.

**Weaknesses:**

1. The paper does not quantitatively discuss how $f_b$ and $\phi$ control the semantic and geometrical information of the classifier. The intuition behind these two functions are not clear enough.

**Questions:**

1. The authors claim (l. 26-26, l. 90-91) that previous works “rely heavily on either simple models or strong assumptions”, but the references seem to be out of date. The authors should include more related work in the past few (2-3) years.
2. It is not proper to claim that the proposed model “explains that properties of TBAs” (l. 39). It seems that this paper only explains those properties of TBA that is mentioned in l.26-35 and l. 82-89. If there are any other properties of TBA that cannot be explained by the proposed model?

---

> ### Author Rebuttal · Authors · 2023-08-10
>
> Dear Reviewer HLto:
>
> Thanks to the reviewer for appreciating our effort and providing helpful feedback. We will clarify the reviewer’s concerns in this response.
>
> **(Weakness 1)**
>
> In this paper, we assume that the source and target model can be decomposed into the product of a semantic classifier $f_b$ and a concentration multiplier $\phi$. In the global response, we further discuss the reasonableness of this assumption. Informally speaking, the intuition behind this assumption is that the classifier can capture both semantic and geometrical information of the natural data (Lines 45-49).
>
> In our analysis, $f_b$ and $\phi$ can be parameterized by the semantic and geometrical information, respectively. Here, the semantic information is represented by subsets of $\mathcal{M}$. Consider the binary classification example. The semantic information of natural data is $A = \{\mathbf{x} : y(\mathbf{x}) = 1\}$ and $B = \{\mathbf{x} : y(\mathbf{x}) = -1\}$ (Line 193), and the semantic information learned by $f_b$ is $A_f$ and $B_f$ (Proposition 4.4). We do not quantitatively discuss the relationship between $A$, $B$, $A_f$, and $B_f$ (i.e., how $f_b$ controls the semantic information of natural data) since our paper only imposes mild assumptions on the manifold $\mathcal{M}$ (i.e., smooth and compact) and the distribution $D$ (i.e., continuous and supported on $\mathcal{M}$). By introducing stronger assumptions on $\mathcal{M}$ and $D$, it is possible to perform quantitative analysis on how $f_b$ controls the semantic information of natural data. However, this analysis is beyond the scope of this paper.
>
> We can do the same analysis on concentration multipliers and geometrical information, which is omitted here.
>
>
> **(Question 1)**
>
> To the best of our knowledge, there has been no theoretical work that explains the transferability of adversarial examples in the past two to three years. This is mainly due to the limited information provided by the black-box attack model.
>
> **(Question 2)**
>
> Apologies for the possible misunderstanding. Our paper mainly discusses the following two properties of TBAs:
> + The adversarial examples of an inaccurate source model can transfer to the target model.
> + The success rates of TBAs are hard to improve.
>
> To the best of our knowledge, the above properties are the only two that satisfy the following criteria:
> + The property commonly held by different methods of TBAs.
> + The property cannot be explained by existing theory.
>
> That is why we only explain these two properties in this paper.
>
> Thanks again for the reviewer’s comments and feedback. We hope that we have addressed the reviewer's concerns. Please let us know if there are any further questions.

---

> > ### Comment · Reviewer_HLto · 2023-08-15
> > **Regarding to the rebuttal**
> >
> > Thanks the authors for addressing our concerns. I would be happy to keep my score and recommand accepting this paper.

---

> > > ### Author Response · Authors · 2023-08-15
> > >
> > > Thanks for the response. We sincerely appreciate that.

---

### Official Review · Reviewer_9Mj1 · 2023-07-01

**Soundness:** 2 fair
**Presentation:** 2 fair
**Contribution:** 4 excellent
**Rating:** 5
**Confidence:** 2

**Summary:**

The paper proposes a new framework, manifold attack models, to explain the transferability of adversarial examples. The model divides adversarial examples into on-manifold examples and off-manifold examples, where the former one lies in the low dimensional manifold of natural data. The paper show that off-manifold adversarial examples are transferable when the source model is inaccurate. It also prove that the non-existence of off-manifold adversarial examples is one of the reasons why the success rates of TBAs are hard to improve.

**Strengths:**

1. The paper takes a large step towards an very important and interesting problem, why adversarial examples are transferable across different models. It conducts  in-depth theoretical analysis into the problem, which previous works did not perform.
2. The theoretical analysis successfully explain find several practical properties of the adversarial examples: (1) an inaccurate source model can transfer to the target model with a non-negligible success rate; (2) the success rates of TBAs are constantly lower than other methods of black-box attacks.

I think the paper makes great contribution towards understanding adversarial examples if the statements are correct. But I cannot judge their correctness based on the current version (see weakness).

**Weaknesses:**

The paper is very hard to read. I suggest the authors to format their presentation of the work in the following aspects.

1. Clearly states the assumption of each proposition and theorem. For example, I think one assumption of propositions in Section 4 is the specific forms of the classifiers, which are not clearly stated.
2. Formally states their propositions and theorems using mathematical languages. For example, replace "suffers from off-manifold adversarial examples" in Proposition 4.6. with formal mathematical equations and explain the meaning of the equations after the theorem.
3. Reduce the number of propositions and give one main theorem for one conclusion with one main theorem.

In addition, it is hard to evaluate the correctness of the these theorems without mathematical languages in current version.

And I think the paper also make strong assumption that the classifier need to have certain forms, which limits the contribution of the paper.

**Questions:**

See Weakness.

**Limitations:**

See Weakness.

---

> ### Author Rebuttal · Authors · 2023-08-09
>
> Dear Reviewer 9Mj1:
>
> Thanks to the reviewer for providing helpful comments and constructive feedback. Apologies for the possible confusion due to the unclear presentation. We will restate and clarify our contributions in our response.
>
> First of all, we would like to clarify that the results from Proposition 4.3 to Proposition 5.9 are rigorously written in mathematical language.
> Our paper seems to include some "mathematically informal" language for the following reason. During the writing of this paper, we intentionally adopt some terminologies (e.g., suffer from, be vulnerable to, transfer to, and be robust against) from previous empirical works, expecting to create some connections between previous works and ours. We will explain the meaning of these terms later in this response. We will also restate the assumptions as required by the reviewer.
>
> As for the concern that this paper provides too many propositions, we will rewrite the propositions that are directly related to our main explanatory results (i.e., Propositions 4.7 and 4.12) as theorems. The propositions that only related to minor results would stay unchanged.
>
> Last but not least, we must clarify that the assumptions in this paper are actually much milder than those in previous theoretical works that study TBA (Lines 90-99). We provide a detailed discussion on the reasonableness of the assumption "the source and target model could be decomposed into the product of a semantic classifier and a concentration multiplier" in the global response.
>
> Thanks again for the reviewer’s comments and feedback. We hope that we have addressed the reviewer's concerns. Please let us know if there are any further questions.
>
> **Restating Terms and Assumptions**
>
> We first restate the major assumptions of our paper.
>
> **Assumption R.1 [cf. Line 120-122]** Let $\mathcal{M} \subset [0,1]^d$ be a compact smooth manifold. Assume that the dimension of $\mathcal{M}$ is less than $d$. Let $D$ be some given continuous distribution that is supported on $\mathcal{M}$. We say that a random variable $x$ is natural data if $x \sim D$.
>
> There is a typo in the definition of adversarial examples in Line 130. In practice, we only care about the adversarial examples of natural data $x \sim D$ instead of any $x \in \mathbb{R}^d$.
>
> **Assumption R.2 [cf. Line 178-184]** Let $f$ be the source or target model of a TBA. If $k=2$, we assume that $\exists f_b, \phi$ such that $f(\mathbf{x}) = f_b(\mathbf{x}) \cdot \phi(\mathbf{x})$ for $\forall \mathbf{x} \in [0,1]^d$. If $k > 2$, we assume that $\exists f_b, \phi$ such that $f^{(j)}(\mathbf{x}) = f_b^{(j)}(\mathbf{x}) \cdot \phi(\mathbf{x}), \forall 1 \leq j \leq k$ for $\forall \mathbf{x} \in [0,1]^d$. Here, $f_b$ and $\phi$ are semantic classifiers and concentration multipliers defined in Definitions 4.1 and 4.3, respectively.
>
> Next, the "mathematically informal" terms are restated as follows.
>
> Given perturbation radius $\delta$ and $x \in \mathbb{R}^d$, a data point $x_a \in B(x;\delta)$ is called an **adversarial example** of $f$ at $x$ if $y(f,x) \neq y(f,x_a)$.
>
> We say that a classifier **suffers from**, or **is vulnerable to** (on-, off-manifold) adversarial examples if $\exists x \sim D, x_a \in B(x;\delta)$ such that $x_a$ is an (on-, off-manifold) adversarial example of $f$ at $x$.
>
> We say that a classifier is **robust against** (on-, off-manifold) adversarial examples if this classifier does not suffer from (on-, off-manifold) adversarial examples.
>
> We say that an adversarial example $x_a$ of $f_s$ at $x$ **transfers to** $f_t$ if $x_a$ is also an adversarial example at $x$ of $f_t$.

---

> ### Author Response · Authors · 2023-08-18
> **We are looking forward to your reply!**
>
> Dear Reviewer 9Mj1:
>
> Thank you again for the effort you put into reviewing our paper. The comments and suggestions are helpful and constructive. We have replied to the proposed concerns and questions in our rebuttal.
>
> We totally understand that this is quite a busy period. We would deeply appreciate it if you could take some time to provide any feedback on whether our responses address your concerns. We would also be glad to reply to any further comments.
>
> Best,
>
> The Authors

---

> ### Comment · Reviewer_9Mj1 · 2023-08-19
> **Post Rebuttal Response**
>
> Thanks for the valuable response for the authors, which partially resolves my concerns. I will increase my score to 5.

---

> > ### Author Response · Authors · 2023-08-19
> >
> > Thanks to the reviewer for reading our rebuttal and raising their score. We sincerely appreciate that.

---

### Official Review · Reviewer_rJF7 · 2023-07-06

**Soundness:** 4 excellent
**Presentation:** 3 good
**Contribution:** 2 fair
**Rating:** 5
**Confidence:** 3

**Summary:**

This paper proposes an explanatory model for the transferability phenomenon of adversarial examples. This model involves a specific hypothesis class that can be decomposed into a “semantic classifier” and a “geometric multiplier.” The authors then use a specific choice of these classifiers to explain two interesting observations in the literature that lacks any theoretical understanding. Finally, the paper also tries to establish a relationship between the proposed model to ReLU neural networks.

---

After reading the rebuttal and the other reviews, I decide to raise the score from 4 to 5.

**Strengths:**

1. This paper provides an easy-to-follow and intuitive model for explaining the transferability of adversarial examples. The decomposition into on- and off-manifold components makes a lot of sense here and is also fairly novel.
2. The paper provides a lot of clear definitions and background which makes it self-contained. I really appreciate the efforts that go into making this paper comprehensive and easy to read.
3. The problem of transferability is indeed not well-explored theoretically. This paper fills in the gap in my understanding fairly well by formalizing many concepts that have been floating around in the literature.

**Weaknesses:**

My main concern about this work is the applicability of the model. The results depend on rather strong assumptions about a specific model, and I am not fully convinced that this model accurately describes a neural network. I list more detailed comments below:

1. All of the results in the paper are based on a specific form of semantic classifier and concentration multiplier from Proposition 4.3 and 4.5, respectively. I wonder if all the results also hold for all combinations of semantic classifiers (Definition 4.1) and concentration multipliers (Definition 4.3).
2. Proposition 4.7, 4.8, and Example 4.10 are shown for a specific choice of the concentration multiplier (i.e., $\phi_{\mathrm{on}}$ or $\phi_{\mathrm{off}}$). This further limits the applicability of these statements. Can the authors justify why these specific choices are reasonable?
3. Proposition 4.6 also seems to hold generally. We can see off-manifold adversarial examples as an $\epsilon$-expansion of the data manifold $\mathcal M$ so if an adversarial example exists in $\mathcal M$, then there must also exist its off-manifold version. Therefore, in my opinion, the major results of the paper start from Proposition 4.7 onward.
4. Section 5.1 shows that the hypothesis class $\mathcal F_{\mathcal M}$ can be approximated with ReLU neural networks, but I believe a reverse statement would be much more powerful. The current result is perhaps not surprising given that it is well-known that a neural network is a universal function approximator. In other words, it is more important for the results in this paper to be relevant in practice if a neural network can be approximated by some $f \in \mathcal F_{\mathcal M}$.
5. I can see that Section 5.2 and Proposition 5.9 attempt to show the universality of the model which partially addresses my first concern. However, I would hope to see a stronger result showing that typical (or even toy-ish) neural networks can indeed be decomposed into the very specific form of semantic classifier and concentration multiplier used in the proofs. It would be powerful evidence for convincing the community that the proposed explanatory model is truly applicable to neural networks. My suggestion is to rely on an empirical experiment on synthetic toy datasets, which have well-defined on- and off-manifold subspaces.
6. It will also be a big plus if the authors can use the developed model to predict some undiscovered empirical behavior of the transferability phenomenon.

### Minor typo

1. Missing parenthesis in Equation (7).

I am happy to reconsider my rating if the authors can address my concerns during the rebuttal period.

**Questions:**

1. Is the following statement from L145-146 contradictory?

    > In most cases, the adversarial perturbations are regarded as imperceptible to humans, which implies that adversarial examples are **not** naturally generated data.

    I can potentially agree that adversarial examples are not in support of $D(\mathbf x)$ because we don’t simply find adversarial examples by sampling from $D(\mathbf x)$. However, the reasoning above does seem contradictory to me.

2. Can the authors justify the need to differentiate the support of $D(\mathbf x)$ from the natural image manifold $\mathcal M$? I am wondering if we can simply unify the two. Here’s my reasoning: based on the paper, my understanding is that $D(\mathbf x)$ represents train/test data of the given classification task (e.g., cats vs dogs), and $\mathcal M$ represents a space of all possible natural images. So the support of $D(\mathbf x)$ is a subset of $\mathcal M$ (please correct me if my understanding is incorrect). Coming back to the cats vs. dogs example, I can see that there is the “noisy” version of the adversarial examples which I’d regard as in **neither** $\mathcal M$ nor the support of $D(\mathbf x)$. Another type of adversarial example might be the **imperceptible** one which I see as something in **both** $\mathcal M$ and support of $D(\mathbf x)$. Now this makes me wonder what are the adversarial examples that are in $\mathcal M$ but **not** on the support of $D(\mathbf x)$? In other words, as far as I know, there are no adversarial examples that look like a snake (or any other objects) in the cats vs. dogs task, for example. So if there are no adversarial examples that fit this last category, can we just unify $\mathcal M$ and the support of $D(\mathbf x)$? What would be the implication in this case?
3. An issue with the assumption on L222 that states

    > we let the semantic classifier of $f_t$ be the classifier $f^*_b$ in Proposition 4.4.

    I interpret this assumption as that the semantic classifier is perfect, and the adversarial examples simply exist because of the geometric multiplier. If I understand correctly, this also implies that $f_t$ does not have on-manifold adversarial examples which I think is not very accurate as the previous work has shown both on- and off-manifold adversarial examples on typical neural networks. Can the authors discuss more the implications of this assumption (e.g., Why is it reasonable? What would fail if we don’t assume this?)?

**Limitations:**

I do not see an explicitly listed limitation section. Please refer to the Weaknesses section.

---

> ### Author Rebuttal · Authors · 2023-08-09
>
> Dear Reviewer rJF7:
>
> Thanks to the reviewer for appreciating our effort and providing constructive feedback. We will reply to the reviewer’s concerns point by point in our response. Due to limited space, we will replace $\mathcal{M}$ and $\mathbf{x}$ by $M$ and $x$.
>
> **Weakness 1**
>
> It is true that the main results are based on a specific form of $f_b$ and $\phi$.
> Definitions 4.1 and 4.2 only provide abstract definitions of $f_b$ and $\phi$ that might better reveal the motivation and intuitions, while specific definitions (e.g., Proposition 4.3 and 4.5) are necessary for quantitative analysis. It is easy to check that $f_b$ and $\phi$ defined by Definitions 4.1 and 4.2 cannot even be parametrized (uniquely decided by the corresponding semantic/geometrical information). Therefore, it is necessary to specify a "specific form" (i.e., Proposition 4.3 and 4.5) of $f_b$ and $\phi$ to obtain quantitative results.
>
> Back to the reviewer’s concern. Since a specific form of $f_b$ and $\phi$ is necessary, our results do not hold for all combinations of $f_b$ and $\phi$ defined by Definition 4.1 and 4.2. However, we can replace the specific forms in Proposition 4.3 and 4.5 with other specific forms. According to the proofs, we believe that any $f_b$ and $\phi$ that is somehow continuous (w.r.t. the distance to semantic/geometrical information) can derive similar results.
>
> **Weakness 2**
>
> In this paper, we only impose mild assumptions on the manifold $M$ (i.e., smooth and compact), the distribution $D$ (i.e., continuous and supported on M), and the geometrical information $G$ (i.e., $G \subset \mathbb{R}^d$ in Proposition 4.5). To perform a more delicate analysis, it is necessary to introduce stronger assumptions on at least one of M, $D$, or $G$. This paper further assumes that $G = M$ in $\phi_{off}$ and $G = \mathcal{N}\_{\delta}(A_f \cup B_f)$ in $\phi_{on}$. Alternatively, by introducing stronger assumptions on $M$ and $D$, it is possible to extend our results to any $\phi$ defined in Proposition 4.5. However, this analysis is beyond the scope of this paper.
>
> Generally speaking, the manifold attack model enables us to construct a variety of different classifiers that have different desirable properties by choosing different $f_b$ and $\phi$. In this paper, we choose the two specific $\phi$ in Equations (6) and (7) for the following reason. Intuitively, we believe that the off-manifold adversarial examples are more transferable than the on-manifold ones. Therefore, the "desirable properties" of our paper is that a classifier suffers from (almost) only either of the on- or off-manifold adversarial examples. We construct such classifiers (by specifying $\phi$ in Equations (6) and (7)) and prove that they are consistent with existing empirical results (cf. Lines 26-36), which provides possible reasons for the properties of TBA. Any other specific choices of $f_b$ or $\phi$ would be good if their combination could also explain the properties. We make our choices because we think they sufficiently solve our problems.
>
> **Weakness 3**
>
> The reviewer’s understanding of $D$ and $M$ may not be correct. In the binary case, the support set of $D$ equals $A \cup B$ (Line 189). The relationship between $A \cup B$ and $M$ is visualized in Figure 1. Apart from natural data, there are also noise vectors in $M$ that lie outside of $A \cup B$. In the cats vs. dogs examples, these noise vectors are more likely to be unrecognizable than look like a snake or any other object, even if they also lie in $M$. On the other hand, any $x \in [0,1]^d$ would be recognizable or even imperceptible to humans if $x$ is close enough to $A \cup B$, regardless of whether $x \in M$ or not.
>
> Informally speaking, we can see the adversarial examples as lying in an expansion of the combination of the semantic and the geometrical information of the classifier, instead of an expansion of $M$. Therefore, Proposition 4.6 does not hold generally by intuition.
>
> **Weakness 4**
>
> The contributions of Section 5.1 are twofold
> 1. The universal approximation theorem has its assumptions and target functions. Approximating functions on a manifold is a stronger result than the universal approximation theorem.
> 2. More importantly, we show that after approximating $f \in F_M$ by ReLU networks, one of the main results in our paper (i.e., Proposition 4.7) can be partially recovered.
>
> Admittedly, a reverse statement would be much more powerful. We discuss the possibility of such approximation in the global response.
>
> **Weakness 5**
>
> See the global response.
>
> **Weakness 6**
>
> Apart from the main explanatory results in Proposition 4.7 and 4.12, our work also proves some minor non-explanatory results. Propositions 4.6 and 4.8 can be viewed as our predictions of the undiscovered empirical behaviors.
>
> **Question 1**
>
> Apologies for the unclear representation. We leave out the assumption of this statement that the adversarial examples should be w.r.t. accurate classifiers and natural data. Assume that an adversarial example is also naturally generated. Since the perturbation is imperceptible, the ture labels (notice that only natural data have ture label) and the predicted labels (of the accurate classifier) of the adversarial example and its corresponding natural data should both be the same, which leads to contradiction.
>
> **Question 2**
>
> We have replied to this question in Weakness 3.
>
> **Question 3**
>
> $f_t = f_b^* \cdot \phi$ does not implies that $f_t$ have no on-manifold adversarial examples. We explain this in the binary case. Since $\phi$ not necessarily learns the exact shape of $M$ (i.e., it is possible for some $x \in M$ that $\phi(x) < 0$), the sign of $f_t$ could be different from $f_b^{*}$, which leads to on-manifold adversarial examples.
>
> Thanks again for the reviewer’s comments and feedback. We hope that we have addressed the reviewer's concerns. Please let us know if there are any further questions.

---

> > ### Comment · Reviewer_rJF7 · 2023-08-14
> > **Response to the rebuttal**
> >
> > I appreciate the efforts the authors have put into putting together the rebuttal and clarifying many of my questions. I might have one follow-up question about the definition of $M$ and the support of $D$ ($A \cup B$).
> >
> > I understand the distinction between $M$ and $A \cup B$ as defined in the paper, but I am not fully convinced that we need to differentiate the two. Put differently, I would define "off-manifold" adversarial examples as $x' \notin A \cup B$ and not rely on $M$ to define the manifold, i.e., just calling $A \cup B$ as "on-manifold" and things outside of it as "off-manifold." I believe this is better aligned with the notion of on- and off-manifold commonly accepted by the community.
> >
> > This is why I asked about the purpose of introducing $M$ in addition to $A \cup B$. One immediate reason I can see is that $M$ is necessary for formalizing the notion of the semantic classifier as one needs support for $A_f$ and $B_f$, but I feel like this is an "artificial" reason to define it. Based on the rebuttal, the authors also do not mean $M$ to represent the distribution of natural images either. So I'm still confused about what $M$ really represents in the real world.
> >
> > Overall, I am satisfied with the authors' clarification, but I believe that my concern about the applicability of the results (i.e., assumptions on the hypothesis class) still holds. That said, I also believe that the contribution of this paper outweighs the limitations. While I am not fully convinced about the usefulness of this result yet, it provides one of the most rigorous analyses of the transferability phenomenon of adversarial examples which is currently lacking. I am confident that the community will benefit from learning about this result and perhaps iterate on it in the future. So I decide to raise my score from 4 to 5 and recommend accepting the paper.

---

> > > ### Author Response · Authors · 2023-08-16
> > >
> > > Thanks to the reviewer for reading our rebuttal and raising their score. We hope that the following response can answer the follow-up question about the definitions of $\mathcal{M}$ and the support of $D$.
> > >
> > > Generally speaking, the main reason for differentiating $\mathcal{M}$ and the support of $D$ is:
> > > > The classification might rely on some geometrical structures of $\mathcal{M}$. The support of $D$ is often not equipped with such geometrical structures.
> > >
> > > For example, consider a binary classification task on a $2$-dimensional swiss roll dataset (e.g., the dataset generated by sklearn.datasets.make\_swiss\_roll; We refer the reviewer to the document of scikit-learn for visualization of such datasets) in $\mathbb{R}^3$. Denote this swiss roll dataset by $\mathcal{M}$ and let the support of $D$ be $A \cup B$, where $A$ and $B$ are subsets of $\mathcal{M}$ with $A \cup B \neq \mathcal{M}$. In this example, $A$ and $B$ could be entangled with each other in $\mathbb{R}^3$ while being separable in $\mathcal{M}$. Therefore, by introducing $\mathcal{M}$ in addition to the support of $D$, it is possible to construct more accurate classifiers.
> > >
> > > It is also necessary to clarify that introducing $\mathcal{M}$ in addition to the support of $D$ (or more specifically, assuming that the support of $D$ is a subset of some manifold $\mathcal{M}$) is not for the purpose of proving our main results. Instead, it is a common assumption that can be found in many previous works ([1], [2], [3], [4], [5]). We can verify this assumption from the following perspectives.
> > > 1. From a generative models perspective. Modern generative models (e.g., GAN [6] and diffusion models [7]) can produce realistic images, which implies that the distribution of natural data is similar (in some sense) to that of the generated data. Meanwhile, theoretical works also prove similar results that generative models can approximate data distributions [8]. On the other hand, Lemma 1 of [9] proves that the image of a fully-connected neural network is contained in a countable union of low-dimensional manifolds. Combining the above statements, it is reasonable to assume that the natural data lie on low-dimensional manifolds.
> > > 2. From a non-linear dimensionality reduction perspective. It is a popular belief that the label of natural data is decided by some latent features ([10], [11]). In practice, non-linear dimensionality reduction would project the high-dimensional natural data onto lower-dimensional manifolds and train the classifier on the projected data without losing much accuracy ([12], [13]). Here, we could informally interpret $\mathcal{M}$ as follows:
> > > > $\mathcal{M}$ represents the (possibly non-linear) space that is spanned by the latent features of the data.
> > >
> > > The above statement answers the reviewer's question of "what $\mathcal{M}$ really represents in the real world".
> > >
> > > Combining the above discussion and the former responses, we believe that we have justified the applicability of our results. Please inform us if there is any further question. We are very happy to clarify any possible confusion.
> > >
> > > **Reference**
> > >
> > > [1] Juan Cervino, Luiz F. O. Chamon, Benjamin David Haeffele, Rene Vidal, Alejandro Ribeiro, Learning Globally Smooth Functions on Manifolds, ICML, 2023
> > >
> > > [2] David Cohen, Tal Shnitzer, Yuval Kluger, Ronen Talmon, Few-Sample Feature Selection via Feature Manifold Learning, ICML, 2023
> > >
> > > [3] Ricardo Dominguez-Olmedo, Amir-Hossein Karimi, Georgios Arvanitidis, Bernhard Schölkopf, On Data Manifolds Entailed by Structural Causal Models, ICML, 2023
> > >
> > > [4] Ilya Kaufman, Omri Azencot, Data Representations’ Study of Latent Image Manifolds, ICML, 2023
> > >
> > > [5] Shikun Sun, Longhui Wei, Junliang Xing, Jia Jia, Qi Tian, SDDM: Score-Decomposed Diffusion Models on Manifolds for Unpaired Image-to-Image Translation, ICML, 2023
> > >
> > > [6] Ian J. Goodfellow, Jean Pouget-Abadie, Mehdi Mirza, Bing Xu, David Warde-Farley, Sherjil Ozair, Aaron C. Courville, and Yoshua Bengio, Generative Adversarial Networks, CoRR abs/1406.2661, 2014.
> > >
> > > [7] Yang Song and Stefano Ermon, Generative Modeling by Estimating Gradients of the Data Distribution, NeurIPS, 2019.
> > >
> > > [8] Minshuo Chen, Wenjing Liao, Hongyuan Zha, and Tuo Zhao, Statistical Guarantees of Generative Adversarial Networks for Distribution Estimation, CoRR abs/2002.03938, 2020.
> > >
> > > [9] Martín Arjovsky and Léon Bottou, Towards Principled Methods for Training Generative Adversarial Networks, ICLR, 2017.
> > >
> > > [10] Zeyuan Allen-Zhu and Yuanzhi Li, Towards Understanding Ensemble, Knowledge Distillation and Self-Distillation in Deep Learning, ICLR, 2023.
> > >
> > > [11] Zeyuan Allen-Zhu and Yuanzhi Li, Feature Purification: How Adversarial Training Performs Robust Deep Learning, FOCS, 2021.
> > >
> > > [12] Guy Rosman, Michael M. Bronstein, Alexander M. Bronstein, Ron Kimmel, Nonlinear Dimensionality Reduction by Topologically Constrained Isometric Embedding. IJCV, 2010
> > >
> > > [13] Christian Walder, Bernhard Schölkopf, Diffeomorphic Dimensionality Reduction, NIPS, 2008.

---

> > > > ### Comment · Reviewer_rJF7 · 2023-08-17
> > > > **Thank you for the extra clarification**
> > > >
> > > > Thank you for answering the questions and also providing more references. This is quite illuminating. I can explain where my question comes from, but this might get philosophical and way off track from your paper. So don't feel pressured to respond at all.
> > > >
> > > > The well-known statement is that high-dimensional data like images live in "a low-dimensional manifold." I understand that $M$ is not the same as support of $D = A \cup B$ from your paper and your example. Given that they are different, my question almost comes down to whether this low-dimensional manifold is represented by $M$ or by the support of $D$. From your description so far, the answer seems to be $M$. One not-so-rigorous way to think about this is when we sample from generative models like GAN or VAE, do we sample from $M$ or $D$ (given that this "low-dimensional manifold" is well-represented by the generative models)? I tentatively think that the answer is $D$ because if you train a GAN on cats ($A$) and dogs ($B$), you sample more cats and dogs ($A \cup B$) and not some kind of interpolation between the two in the *ideal* case. But one could certainly argue that interpolating in the latent space of GAN or VAE does lead to some kind of interpolation in the output space which would support the answer being $M$. But then again, having the "low-dimensional manifold" statement tied to certain models does not seem very satisfying to me. Perhaps, all of this will become clear once I check out the papers you linked!

---

> > > > > ### Author Response · Authors · 2023-08-18
> > > > >
> > > > > Thank you for your response. We interpret the reviewer’s question as “why the low-dimensional manifold is represented by $\mathcal{M}$ instead of $D = A \cup B$”. In this response, we only briefly explain our reason from the generative models perspective since this question is indeed way off track from our paper.
> > > > >
> > > > > Mathematically, a generative model (e.g., GAN or VAE) can be formally defined as a mapping from an easy-to-sample distribution [1], e.g., uniform and Gaussian distribution, to the fake distribution that approximates $D$. In the ideal case, the generative model $G$ takes, for example, a Gaussian random variable $\mathbf{x} \sim \Phi$ as input and outputs a cat (or dog) image $G(\mathbf{x}) \in A \cup B$. However, the output could be unrecognizable if the input $\mathbf{x}$ is not Gaussian, i.e., $ G(\mathbf{x}) \in \mathcal{M}$ in the general case. That is why choosing $\mathcal{M}$ to represent the underlying manifold is more reasonable than choosing $D$.
> > > > >
> > > > > We hope that the above discussion would answer the reviewer’s question. Please inform us if there is any further question. We are very happy to clarify any possible confusion.
> > > > >
> > > > > **Reference**
> > > > >
> > > > > [1] Minshuo Chen, Wenjing Liao, Hongyuan Zha, and Tuo Zhao, Statistical Guarantees of Generative Adversarial Networks for Distribution Estimation, CoRR abs/2002.03938, 2020.

---

### Official Review · Reviewer_KVUg · 2023-07-12

**Soundness:** 3 good
**Presentation:** 2 fair
**Contribution:** 2 fair
**Rating:** 4
**Confidence:** 3

**Summary:**

This paper provides a analysis of transfer-based attacks within a unified theoretical framework. The authors introduce the manifold attack model, which serves as an explanatory tool, formalizing commonly held beliefs and shedding light on previously observed empirical results. Through their model, they elucidate the phenomenon of transferability in adversarial examples, even when the source model is inaccurate, by highlighting the role of the data manifold's curvature.

**Strengths:**

The paper addresses the crucial problem of transfer-based attacks, which are a significant threat to the security and robustness of machine learning models.

**Weaknesses:**

One major concern with this paper is the lack of sufficient quantitative analyses. While the proposed manifold attack model offers a theoretical framework for understanding transfer-based attacks, it would greatly benefit from concrete quantitative analyses to support its claims. One suggestion is to fit the model using low-dimensional data and perform a thorough verification. This would not only provide empirical evidence for the effectiveness of the proposed model but also enhance the paper's credibility by demonstrating its applicability in practical scenarios.

**Questions:**

None

**Limitations:**

Yes

---

> ### Author Rebuttal · Authors · 2023-08-09
>
> Dear Reviewer KVUg:
>
> Thank the reviewer for providing helpful comments and constructive feedback. We will clarify the possible confusion in our response.
>
> As its title suggests, our paper's primary effort is to explain the properties of TBAs from a theoretical perspective. We propose an explanatory model that is consistent with existing results. In other words, the existing quantitative analyses (e.g., Dong et al., 2022; Papernot et al., 2017; Tramèr et al., 2017) could already support our claims. Besides, we also provide a detailed discussion in Appendix A on the reason why further empirical verification is unnecessary for validating our proposed model.
>
> However, we are indeed inspired by the reviewer's suggestion. We fit the proposed model using (synthetic) low-dimensional data and conduct numerical experiments on neural networks. A summary of the experiments and the corresponding analysis can be found in the global response.
>
> Thanks again for the reviewer’s comments and feedback. We hope that we have addressed the reviewer's concerns. Please let us know if there are any further questions.
>
> **References**
>
> Yinpeng Dong, Shuyu Cheng, Tianyu Pang, Hang Su, and Jun Zhu. Query-efficient black-box adversarial attacks guided by a transfer-based prior. IEEE Trans. Pattern Anal. Mach. Intell., 2022.
>
> Nicolas Papernot, Patrick D. McDaniel, Ian J. Goodfellow, Somesh Jha, Z. Berkay Celik, and Ananthram Swami. Practical black-box attacks against machine learning. In AsiaCCS, 2017.
>
> Florian Tramèr, Nicolas Papernot, Ian J. Goodfellow, Dan Boneh, and Patrick D. McDaniel.  The space of transferable adversarial examples. CoRR, abs/1704.03453, 2017.

---

> ### Author Response · Authors · 2023-08-18
> **We are looking forward to your reply!**
>
> Dear Reviewer KVUg:
>
> Thank you again for the effort you put into reviewing our paper. The comments and suggestions are helpful and constructive. We have replied to the proposed concerns and questions in our rebuttal.
>
> We totally understand that this is quite a busy period. We would deeply appreciate it if you could take some time to provide any feedback on whether our responses address your concerns. We would also be glad to reply to any further comments.
>
> Best,
>
> The Authors

---

### Author Rebuttal · Authors · 2023-08-10

Dear Reviewers:

Thanks to the reviewers for their helpful comments and constructive feedback. We adopt the suggestion of verifying our assumption from multiple reviews. In this response, we further discuss the reasonableness of our assumption and provide a brief summary of the additional numerical experiment.

The reviewers' major concern is about the reasonableness of the assumption that is formalized as follows:

**Assumption R.2 [cf. Line 178-184]** Let $f$ be the source or target model of a TBA. If $k=2$, we assume that $\exists f_b, \phi$ such that $f(\mathbf{x}) = f_b(\mathbf{x}) \cdot \phi(\mathbf{x})$ for $\forall \mathbf{x} \in [0,1]^d$. If $k > 2$, we assume that $\exists f_b, \phi$ such that $f^{(j)}(\mathbf{x}) = f_b^{(j)}(\mathbf{x}) \cdot \phi(\mathbf{x}), \forall 1 \leq j \leq k$ for $\forall \mathbf{x} \in [0,1]^d$. Here, $f_b$ and $\phi$ are semantic classifiers and concentration multipliers defined in Definitions 4.1 and 4.3, respectively.

Assumption R.2 suggests that a neural network can be exactly (or approximately) decomposed into the product of two functions of specific forms. As is commented by Reviewer rJF7, such results can be viewed as the "reverse of the universal approximation theorems". To the best of our knowledge, there are no existing theoretical or empirical results that prove similar results.

Here, we take a step back and consider the following weaker version of Assumption R.2. For the sake of simplicity, we only consider the binary classification in Euclidean space (cf. Lines 246-252). Recall that given compact smooth manifold $\mathcal{M}$, exists $\Delta > 0$ such that $\mathcal{N}_{\Delta}\mathcal{M}$ is a tubular neighborhood of $\mathcal{M}$ (cf. Lemma 4.12).

**Assumption R.3** Let $f$ be the source or target model of a TBA. Let $\mathcal{M}\_1$ be a compact smooth manifold that is not necessarily a subset of $\mathcal{M}$. Let $\mathcal{N}\_{\Delta\_1}\mathcal{M}\_1$ be a tubular neighborhood of $\mathcal{M}\_1$ for some $\Delta_1 > 0$. We assume that for $\forall \mathbf{x} \in \mathcal{M}\_1$ and for any non-zero $ \mathbf{u} \in N\_{\mathbf{x}} \mathcal{M}\_1$, we have: $f(\mathbf{x} + r_1 \mathbf{u}) > f(\mathbf{x} + r_2 \mathbf{u})$ holds for $\forall \Delta > r_2 > r_1 > 0$.

By letting $G$ in Definition 4.2 take $G = \mathcal{M}\_1$, the classifier in Assumption R.2 would also satisfy Assumption R.3. Next, we verify this assumption on low-dimensional linearly separated data. Given $d > \kappa > 0$, consider the $\kappa$-dimensional manifold (in this case, a flat plane) $\mathcal{M} = \\{ (x_1, x_2, \cdots, x_{\kappa -1}, x_{\kappa} , 0, \cdots, 0) : x_i \in \mathbb{R}, \forall i = 1, 2, \cdots, \kappa \\} \subset \mathbb{R}^d$.
Assume that the natural data $\mathbf{x} \in \mathcal{M}$ is drawn from some continuous distribution $D(\mathbf{x})$, say, a mixture of Gaussian distribution.

For the sake of simplicity, let the natural data be linearly separable in the following sense.
Given $\lambda > 0$ and natural data $\mathbf{x} = (x_1,\cdots, x_d)$, we assume that $|x_1| > \lambda$, and the label of $\mathbf{x}$ is decided by $y(\mathbf{x}) = 1$ if $x_1 > \lambda$, and $y(\mathbf{x}) = -1$ if $x_1 < -\lambda$.

Our goal is to train a classifier that satisfies Assumption R.3.
Here, we briefly introduce the (theoretical) training settings.
Given $N > 0$, let the training set $S_N = \\{ \mathbf{x}_1, \mathbf{x}_2, \cdots, \mathbf{x}_N \\} \sim D^N(\mathbf{x})$ be independent and identically distributed (i.i.d.) random variables.
For $\forall 1 \leq n \leq  N$, denote the label of $\mathbf{x}_n$ by $y_N$.
Since $D$ is continuous, we assume without loss of generality that $\left( \exists a \in \mathbb{R} \ s.t. \ \mathbf{x}_i = a \mathbf{x}_j \right) \implies i=j$ holds for $\forall 1 \leq i, j \leq N$.

The hypothesis class is defined as $\mathcal{F}\_{\mathbf{w}} = \\{ f\_{\mathbf{w}}: \mathbf{w} \in \mathbb{R}^d \\}$, where $f\_{\mathbf{w}}(\mathbf{x}) = \langle \mathbf{w}, \frac{\mathbf{x}}{ || \mathbf{x} ||\_2}  \rangle$ for $\forall \mathbf{x} \in \mathbb{R}^d$.
Consider the logistic loss function $\ell(u) := \log (1 + \exp(-u)),\ \forall u \in \mathbb{R}$, and the gradient descent (GD) iteration $\mathbf{w}(t+1) = \mathbf{w}(t) - \eta  \nabla_{\mathbf{w}} \mathcal{L}_N(\mathbf{w}(t))$.
We can show that (formal statements and proofs are not provided in this response due to limited space):
+ The GD iteration converges to an accurate classifier $\hat{f}$,
+ $\exists \mathcal{M}_1$ such that $\hat{f}$ satisfy Assumption R.3,

which implies that **we can actually train an accurate classifier (instead of constructing one using oracle information) that satisfies our assumption.**

We also validate this assumption on neural networks based on numerical experiments. We modify the above theoretical training setting by:
+ In order to visualize our results, the manifold $\mathcal{M}$ is set to be a 2-dimensional plane embedded in $\mathbb{R}^3$.
+ The distribution is a mixture of two Gaussian distributions with means $(2,2,0)$ and $(-2,-2,0)$ and covariance $\sigma = 0.5$.
+ The loss function is the binary cross-entropy loss (i.e., the logistic loss).
+ The optimizer is SGD.

We train a two-layer network, also denoted by $\hat{f}$, that reaches 100% testing accuracy.
We check whether $\hat{f}$ satisfies Assumption R.3.
The corresponding experimental results are presented in the attached PDF file.
The experimental results can be interpreted as follows. Denote the vector $\mathbf{x} \in \mathbb{R}^3$ by $\mathbf{x} = (x,y,z)$. The bottom plane is the manifold, i.e., the $(x,y)$-plane. For each sub-figure, we consider the score of $\hat{f}$ at $(x,y,z)$ for a specific $z$ chosen from $\\{0, 3,5,7 \\}$. The height of the surface is the score of $\hat{f}$. Obviously, the score of $\hat{f}$ decreases as $z$ goes larger, which in accord with the claim in Assumption R.3.

---

### Decision · Program_Chairs · 2023-09-21

**Decision:**

Accept (poster)

**Comment:**

This paper proposes a unified theoretical framework to explain the properties of transferable black-box attacks. To this end, the authors introduced an explanatory model called the manifold attack model, and showed that the transferable adversarial examples exist off-manifold and provide theoretical explanations for common beliefs in the field.

The paper initially received two Strong Accept and three Borderline Rejects. The critical concerns raised by the reviewers were about the soundness of the assumptions such as the generality of the analysis based on the specific form of the classifier, and the lack of quantitive analysis. The authors provided comprehensive responses in the rebuttal, most of which successfully resolved the concerns. As a result, the reviewers updated their recommendations to two strong accept, two borderline accept, and one borderline reject. The negative reviewer however did not respond to the rebuttal.

After reading the paper and discussions, AC agrees with the reviewers that the theoretical explanation of the paper is sound and novel, and hence recommends acceptance. The authors should update the camera-ready paper to include the changes suggested by the reviewers during the review process.